# Molecular bases for HOIPINs-mediated inhibition of LUBAC and innate immune responses

Daisuke Oikawa[1], Yusuke Sato[2,3,4], Fumiaki Ohtake[5,6], Keidai Komakura[1,7], Kazuki Hanada[8], Koji Sugawara[7], Seigo Terawaki[1], Yukari Mizukami[7], Hoang T. Phuong[7], Kiyosei Iio[9], Shingo Obika[9], Masaya Fukushi[10], Takashi Irie[10], Daisuke Tsuruta[7], Shinji Sakamoto[8], Keiji Tanaka[5], Yasushi Saeki[5], Shuya Fukai[2,3] & Fuminori Tokunaga[1✉]

The NF-κB and interferon antiviral signaling pathways play pivotal roles in inflammatory and innate immune responses. The LUBAC ubiquitin ligase complex, composed of the HOIP, HOIL-1L, and SHARPIN subunits, activates the canonical NF-κB pathway through Met1-linked linear ubiquitination. We identified small-molecule chemical inhibitors of LUBAC, HOIPIN-1 and HOIPIN-8. Here we show that HOIPINs down-regulate not only the proinflammatory cytokine-induced canonical NF-κB pathway, but also various pathogen-associated molecular pattern-induced antiviral pathways. Structural analyses indicated that HOIPINs inhibit the RING-HECT-hybrid reaction in HOIP by modifying the active Cys885, and residues in the C-terminal LDD domain, such as Arg935 and Asp936, facilitate the binding of HOIPINs to LUBAC. HOIPINs effectively induce cell death in activated B cell-like diffuse large B cell lymphoma cells, and alleviate imiquimod-induced psoriasis in model mice. These results reveal the molecular and cellular bases of LUBAC inhibition by HOIPINs, and demonstrate their potential therapeutic uses.

[1] Department of Pathobiochemistry, Graduate School of Medicine, Osaka City University, Osaka, Japan. [2] Institute for Quantitative Biosciences, The University of Tokyo, Tokyo, Japan. [3] Synchrotron Radiation Research Organization, The University of Tokyo, Tokyo, Japan. [4] Center for Research on Green Sustainable Chemistry, Tottori University, Tottori, Japan. [5] Laboratory of Protein Metabolism, Tokyo Metropolitan Institute of Medical Science, Tokyo, Japan. [6] Institute for Advanced Life Sciences, Hoshi University, Tokyo, Japan. [7] Department of Dermatology, Graduate School of Medicine, Osaka City University, Osaka, Japan. [8] Pharmaceutical Frontier Research Laboratories, Central Pharmaceutical Research Institute, JT Inc., Kanagawa, Japan. [9] Chemical Research Laboratories, Central Pharmaceutical Research Institute, JT Inc., Osaka, Japan. [10] Department of Virology, Graduate School of Biomedical and Health Sciences, Hiroshima University, Hiroshima, Japan. ✉email: ftokunaga@med.osaka-cu.ac.jp

The pathogen-associated molecular patterns (PAMPs) predominantly activate the nuclear factor-κB (NF-κB) and type I interferon (IFN) antiviral signaling pathways[1]. Various post-translational modifications, such as phosphorylation and ubiquitination, regulate these signaling pathways[2]. The ubiquitin system, composed of ubiquitin-activating enzyme (E1), ubiquitin-conjugating enzyme (E2), and ubiquitin ligase (E3), regulates numerous cellular functions through the generation of various ubiquitin chains in a system referred to as the ubiquitin code[3,4]. The E3s are crucial for substrate recognition, and they are classified into the homologous to the E6AP carboxyl terminus (HECT)-type, the really interesting new gene (RING)-type, and the RING-between-RING (RBR)-type[5,6].

The linear ubiquitin chain assembly complex (LUBAC), composed of the HOIL-1L (also known as RBCK1), HOIP (RNF31), and SHARPIN, specifically generates the Met1-linked linear polyubiquitin chain[7,8]. Among them, HOIP contains the E3 active site. HOIP, as well as HOIL-1L, belongs to the RBR-type E3 family, and the family E3s reportedly generate polyubiquitin chains through a RING-HECT-hybrid reaction[6]. During the linear ubiquitination, the RING1 domain in HOIP binds a ubiquitin-charged E2. Subsequently, the donor ubiquitin is transiently transferred to the active site Cys885 in the RING2 domain via a thioester linkage. Finally, the donor ubiquitin is conjugated to an acceptor ubiquitin, which is captured in the C-terminal linear ubiquitin chain determining domain (LDD) of HOIP, to specifically generate a linear ubiquitin chain[9–12].

Upon stimulation by proinflammatory cytokines, such as tumor necrosis factor-α (TNF-α) and interleukin-1β (IL-1β), LUBAC is recruited to cytokine receptors through binding to K63-linked polyubiquitin chains[13], and conjugates linear ubiquitin chains to NEMO, RIP1, and FADD[14–16]. The linear ubiquitin chain functions as a scaffold to recruit other IκB kinase (IKK) molecules, and the recruited IKK molecules are then activated by a trans-phosphorylation mechanism, leading to the activation of the NF-κB signaling pathway[17]. At present, LUBAC in known to participate in several canonical NF-κB signaling pathways induced by stimulation with the proinflammatory cytokine, PAMPs, T cell receptor agonist, genotoxic stress, and NOD2-mediated pathways[7,8]. However, LUBAC is not involved in either the B cell receptor-mediated pathway or the non-canonical NF-κB pathway[18,19]. Interestingly, genetic mutations of LUBAC subunits and linear ubiquitin-binding proteins induce various disorders[20].

LUBAC inhibitors will facilitate investigations of the enzymatic mechanisms of LUBAC and NF-κB signaling, and will also be useful therapeutics for NF-κB-related disorders. At present, BAY11-7082[21], gliotoxin[22], peptidyl inhibitors of LUBAC[23–25], bendamustine[26], and α,β-unsaturated methyl ester-containing compounds[27] reportedly inhibit the LUBAC activity. We identified a thiol-reactive, α,β-unsaturated carbonyl-containing chemical compound, named HOIPIN-1 from HOIP inhibitor-1, as a LUBAC inhibitor[28]. Furthermore, we developed seven derivatives of HOIPIN-1, and found that HOIPIN-8 is the most potent LUBAC inhibitor among them[29]. However, the detailed molecular mechanism and the pharmacological effects of HOIPINs have remained elusive. Here we investigated the biochemical mechanism of HOIPINs on LUBAC, the cellular effects on the innate immune responses, and the potential therapeutic targets.

## Results

### HOIPINs suppress cytokine-induced NF-κB activation.

We reported that HOIPIN-1 and HOIPIN-8 (Fig. 1a) suppress the LUBAC-induced linear ubiquitination and NF-κB target gene expression[28,29]. Among reported LUBAC inhibitors, BAY11-7082

and gliotoxin showed potent cytotoxicity (Supplementary Fig. 1a). Moreover, these compounds did not suppress the LUBAC-induced NF-κB luciferase activity and the intracellular linear ubiquitin levels at concentrations under the non-toxic levels (Supplementary Fig. 1b, c). Although a cytotoxic dose of gliotoxin suppressed the IL-1β-stimulated NF-κB activation in A549 cells, it further suppressed the activation of JNK, a MAP kinase (MAPK) (Supplementary Fig. 1d). Thus, these compounds do not seem to be LUBAC-specific inhibitors.

To elucidate the inhibitory effects of HOIPINs on the LUBAC-mediated NF-κB signaling, we first analyzed the proinflammatory cytokine-induced NF-κB activation in A549 cells. Then, the amounts of IL-1β-induced intracellular linear ubiquitin were reduced, without affecting the expression levels of the endogenous LUBAC subunits, in HOIPIN-1 and HOIPIN-8-treated cells (Fig. 1b). Moreover, the IL-1β-induced NF-κB activation, demonstrated by the phosphorylation of IKKα/β, p105, and p65, was suppressed by HOIPINs. As reported[29], the inhibitory effects of HOIPIN-8 were more potent than those of HOIPIN-1. Similarly, HOIPIN-1 dose-dependently suppressed TNF-α-induced NF-κB activation and gene expression in A549 cells (Supplementary Fig. 2a, b). In contrast, HOIPINs showed no effects on the activation of JNK (Fig. 1b, Supplementary Fig. 2a) and TAK1 (Supplementary Fig. 2c), suggesting that HOIPINs do not affect the MAP kinase pathway. Upon TNF-α stimulation, the components of the NF-κB pathway are recruited to TNF receptor 1 (TNFR) to form the TNFR signaling complex I[30]. In the presence of HOIPIN-1, dose-dependent decrease in the phosphorylation of p105 and p65 was detected in the lysates (Fig. 1c). Although the amounts of linear ubiquitin in complex I were reduced in the presence of HOIPIN-1, the polyubiquitination of RIP1 and the recruitment of the LUBAC subunits to TNFR were not affected. Moreover, the TNF-α-induced K63-linked ubiquitination of RIP1 was not largely down-regulated by HOIPIN-1 (Supplementary Fig. 2d). In contrast, the linear ubiquitination of endogenous NEMO and the subsequent activation of the canonical IKK were dose-dependently suppressed in the presence of HOIPIN-1 (Fig. 1d, e). Indeed, the TNF-α-induced intra-nuclear translocation of p65 was retarded in HOIPIN-1-treated cells (Fig. 1f), and the secretion of IL-6 was reduced in IL-1β-treated A549 cells (Supplementary Fig. 2e). To examine the effects of HOIPINs on IL-1β-induced gene expression, we performed an unbiased RNA sequencing (RNA-seq) analysis. A principal component analysis revealed that the HOIPINs exerted minimal effects on the transcription in A549 cells in the absence of IL-1β (Fig. 1g). Although IL-1β drastically modulated the gene expression, the HOIPINs strongly suppressed the IL-1β-induced gene expression. Indeed, the expression of NF-κB target genes was canceled in the presence of HOIPINs (Fig. 1h, i). We further examined the effect of HOIPIN-8 on TNF-α-induced gene expression by RNA-seq (Supplementary Fig. 2f–k). The TNF-α treatment of A549 cells enhanced the expression of multiple genes involved in several pathways, affecting inflammatory signaling, infectious disorders, and cellular processes (Supplementary Fig. 2h). HOIPIN-8 showed suppressive effects on these pathways, but minimally affected the necroptosis pathway (Supplementary Fig. 2i–k).

We previously reported the enhanced NF-κB activation in optineurin-deficient HeLa cells[31]. HOIPINs suppressed the elevated NF-κB activity in IL-1β-treated optineurin-deficient cells (Supplementary Fig. 3a–d). Furthermore, HOIPINs suppressed TNF-α-mediated NF-κB activation and target gene expression in Jurkat cells, a human T cell line, whereas it exerted minimal effects on HOIP-deficient Jurkat cells (Supplementary Fig. 3e–g), suggesting that LUBAC is the major target of HOIPINs in the suppression of the NF-κB pathway. Collectively, these results

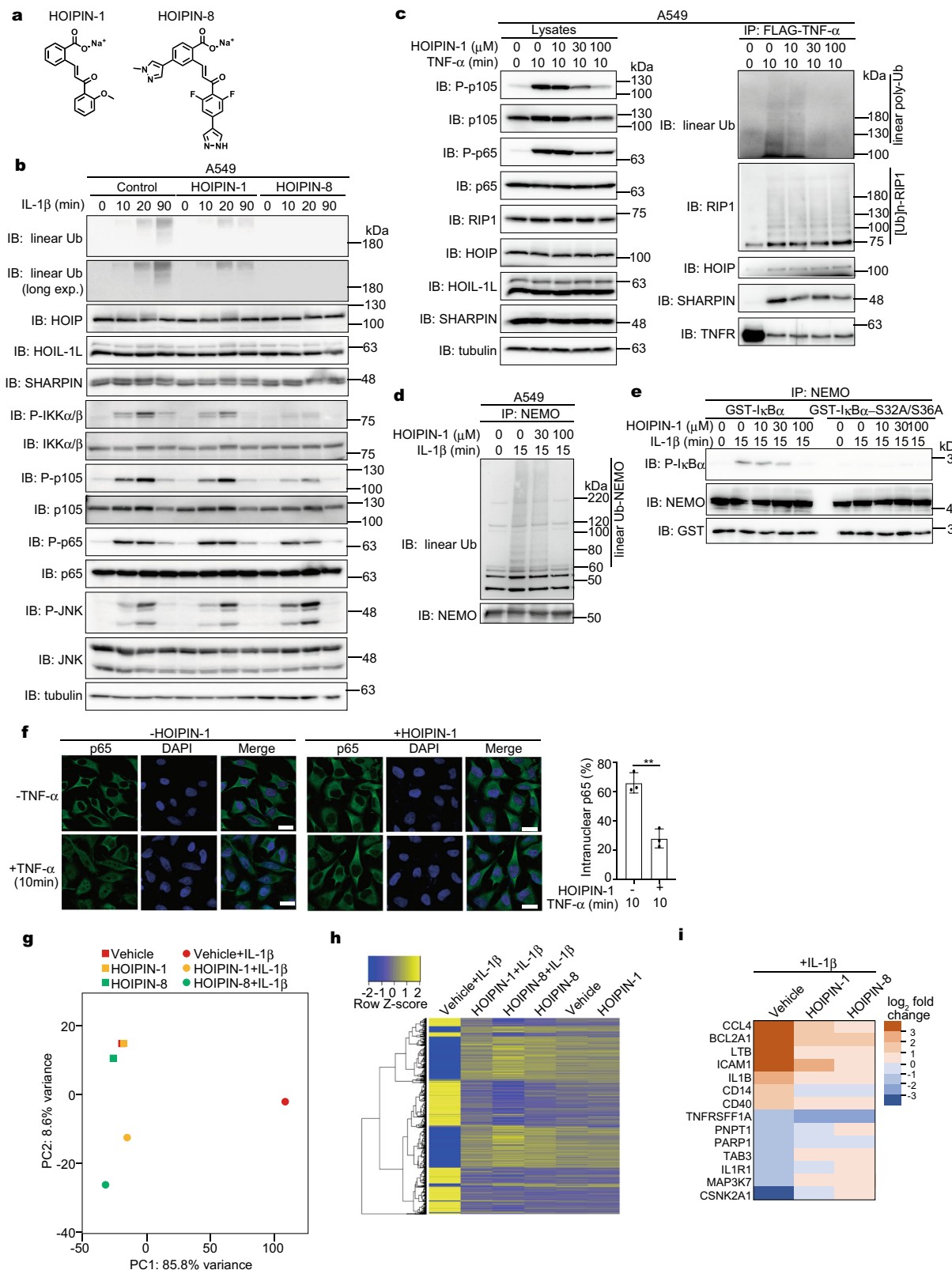

indicate that HOIPINs predominantly inhibit the LUBAC-mediated canonical NF-κB activation induced by proinflammatory cytokines.

**HOIPINs suppress LUBAC-associated innate immune responses**. LUBAC is reportedly further involved in the PAMPs-

and T cell receptor-mediated NF-κB activation pathway. However, it does not participate in the non-canonical and B cell receptor-mediated NF-κB pathways[7,8]. We confirmed that HOIPIN-1 inhibited the CD40-mediated NF-κB activation (Supplementary Fig. 4a–c), but not the B cell receptor-mediated pathway (Supplementary Fig. 4d). In contrast, the T cell receptor-mediated NF-κB activation in mouse splenic T cells was

**Fig. 1 HOIPINs inhibit inflammatory cytokine-induced NF-κB activation. a** Chemical structures of HOIPIN-1 and HOIPIN-8. **b** IL-1β-induced NF-κB activation are suppressed by HOIPINs. A549 cells were pre-treated with 30 μM HOIPIN-1 or HOIPIN-8 for 30 min, and then stimulated with 1 ng/ml IL-1β with HOIPINs for the indicated period. After SDS-PAGE, cell lysates were immunoblotted with the indicated antibodies. **c** TNFR complex I formation is not suppressed by HOIPIN-1. A549 cells were pre-treated with the indicated concentrations of HOIPIN-1 for 30 min, and stimulated with 1 μg/ml FLAG-TNF-α and HOIPIN-1 for 10 min. Cell lysates and anti-FLAG immunoprecipitates were immunoblotted with the indicated antibodies. **d** Suppression of IL-1β-induced linear ubiquitination of NEMO in the presence of HOIPIN-1. A549 cells were pre-treated with the indicated concentrations of HOIPIN-1 for 3 h, and stimulated with 1 ng/ml IL-1β with HOIPIN-1. After heat denaturation, the endogenous NEMO was immunoprecipitated and immunoblotted with the indicated antibodies. **e** Reduced canonical IKK activity by HOIPIN-1. A549 cells were pre-treated with the indicated concentrations of HOIPIN-1, and stimulated with 1 ng/ml IL-1β and HOIPIN-1 for 15 min, and then NEMO was immunoprecipitated. The in vitro IKK activity was assessed using GST-IκBα or its Ser32 → Ala/Ser36 → Ala mutant as substrates. **f** TNF-α-induced nuclear translocation of p65 is retarded by HOIPIN-1. HeLa cells were pre-treated with or without 100 μM HOIPIN-1 for 30 min, and stimulated with 20 ng/ml TNF-α for 10 min. Immunofluorescent staining of p65 and nuclear staining were analyzed, and intranuclear p65 was counted. Bars, 20 μm. Data are shown as mean ± SEM, $n = 3$, NS not significant, **$P < 0.01$, Student's $t$-test. **g–i** The IL-1β-induced gene expression in A549 cells was canceled by HOIPINs. A549 cells were pre-treated with 100 μM HOIPIN-1 or 30 μM HOIPIN-8 for 2 h, and stimulated with 1 ng/ml IL-1β for 2 h. The cells were lysed, and subjected to transcriptome-wide expression using their extracted total RNA. A principal component analysis of the RNA-seq analysis (**g**), the heatmaps of the gene expression in the inflammatory pathway (**h**), and the major genes in the NF-κB signal affected by HOIPINs (**i**) are shown.

suppressed by HOIPIN-1 (Supplementary Fig. 4e). Moreover, HOIPIN-1 had little inhibitory effect on the lymphotoxin-β (LT-β)-induced intranuclear translocation of p52, a characteristic of the non-canonical NF-κB activation pathway (Supplementary Fig. 4f), indicating that the inhibitory effects of HOIPIN-1 agree well with the spectrum of the LUBAC-associated NF-κB activation pathways.

Various PAMPs activate the Toll-like receptors (TLRs)-mediated NF-κB and IFN antiviral pathways[1]. Although LUBAC activity reportedly down-regulate the type I IFN production pathway[32,33], recent studies suggested that the LUBAC is necessary for the TLR-mediated interferon regulatory factor 3 (IRF3) activation[34–36]. Among the PAMPs, HOIPIN-1 suppressed the NF-κB activation induced by CpG, a ligand for TLR9 (ref. [37]) (Supplementary Fig. 5a, b). Upon stimulation with lipopolysaccharide (LPS), a TLR4 ligand[37], the phosphorylation of NF-κB and IFN signaling factors, such as p100, p65, and IRF3, was suppressed by HOIPINs in mouse bone marrow-derived macrophages (BMDM) (Fig. 2a, Supplementary Fig. 5c). In addition to NF-κB target genes, the mRNA and protein levels of IRF3 targets were suppressed in HOIPINs-treated BMDM and BJAB cells after LPS stimulation (Fig. 2b, c, Supplementary Fig. 5d, e). Moreover, the TLR3 ligand poly(I:C)-induced phosphorylation of TANK-binding kinase 1 (TBK1) and IRF3 was suppressed by HOIPIN-8 in mouse embryonic fibroblasts (MEF) (Fig. 2d), and the expression of IRF3-target genes was also inhibited by HOIPINs in BMDM and MEF cells (Fig. 2e, Supplementary Fig. 5f). Attenuation of the IFN production pathway by HOIPIN-1 was confirmed by an ISRE-luciferase reporter assay in poly(I:C)-stimulated MEF cells (Fig. 2f). Furthermore, the upregulated phosphorylation of IRF3 and TBK1 and the induction of IRF3-target genes were suppressed by HOIPINs in poly(I:C)-treated WT-MEF cells (Fig. 2g, h). In contrast, in *HOIP^{−/−}*-MEF cells, the poly(I:C)-induced phosphorylation of IRF3 and the expression of IRF3-target genes were negligible. We also found that the Sendai virus (SeV)-induced expression of IRF3-target genes was suppressed by HOIPINs in MEF cells (Fig. 2i). Collectively, these results indicated that the LUBAC activity is required for the activation of the antiviral pathway, and HOIPINs inhibit the innate immune responses induced by various PAMPs or viral infection.

To further investigate the LUBAC-mediated activation of the IFN pathway, we performed a luciferase reporter assay with the promoters of IFNβ (including the IRF3 and NF-κB cis-elements), NF-κB, and Ifit1 (containing only the IRF3 cis-elements). Although the expression of LUBAC enhances the NF-κB activity, it did not enhance the IFNβ and Ifit1 reporter activities. When

LUBAC was co-expressed with IRF3, all the reporter activities were enhanced, and HOIPIN-8 suppressed these activities (Supplementary Fig. 6a), suggesting that HOIPIN-8 exerts inhibitory effects on the IRF-mediated signaling pathway. Co-expression of the active site mutant (C885A) of HOIP with HOIL-1L and SHARPIN, designated as LUBAC-C885A, failed to induce IFNβ- and NF-κB luciferase activities, and showed no increase in the Ifit1 activity with IRF3. In the presence of the constitutively active mutant of IRF3 (IRF3-5D)[38], partial and severe suppressions of the IFNβ and NF-κB activities, respectively, were detected in the presence of HOIPIN-8 (Supplementary Fig. 6a). Moreover, the LUBAC activity was required for the intranuclear translocation of phospho-IRF3 (Supplementary Fig. 6b), and the expression of the active-site mutant of LUBAC suppressed the poly(I:C)-induced TBK1 activation in a dominant-negative manner (Supplementary Fig. 6c). Taken together, these results indicate that the LUBAC is crucial for the IFN production pathway induced by various PAMPs. HOIPINs down-regulate the LUBAC-mediated TBK1 activation, and then attenuate the antiviral responses.

**HOIPINs inhibit the RING-HECT-hybrid reaction of LUBAC.** To investigate the selectivity of HOIPINs in linear ubiquitination, we first performed immunoblotting analyses using cell lysates of A549 cells and BMDM treated with TNF-α, IL-1β, or poly(I:C) (Fig. 3a). The intracellular amounts of linear ubiquitin were increased with these stimulations, but suppressed in the presence of HOIPIN-8. In contrast, immunoblotting with anti-K63-, K48-, and pan-ubiquitin antibodies revealed almost no effect by HOIPIN-8. Furthermore, we quantified the intracellular ubiquitin linkages by mass spectrometry (Supplementary Fig. 7a). The contents of linear ubiquitin were upregulated by stimulation, and the upregulation was canceled in the presence of HOIPIN-8. In contrast, the intracellular contents of the K11-, K29, K48-, and K63-linked ubiquitin chains, the dominant intracellular Lys-linked ubiquitin linkages, were not affected by HOIPIN-8. These results clearly indicated that HOIPINs specifically down-regulate linear ubiquitination, but not Lys-linked ubiquitin chains.

We further confirmed that the in vitro linear ubiquitination activity of the recombinant full-length HOIL-1L/HOIP complex was dose-dependently suppressed by HOIPIN-1 (Fig. 3b). In contrast, HOIPINs showed little inhibitory effect on the in vitro E3 activities of c-IAP2 (RING family), E6AP (HECT family), or HHARI (RBR family) (Supplementary Fig. 7b, c, Fig. 3c). Moreover, the parkin-mediated ubiquitination and degradation of a mitochondrial protein, mitofusin-1, were not affected by

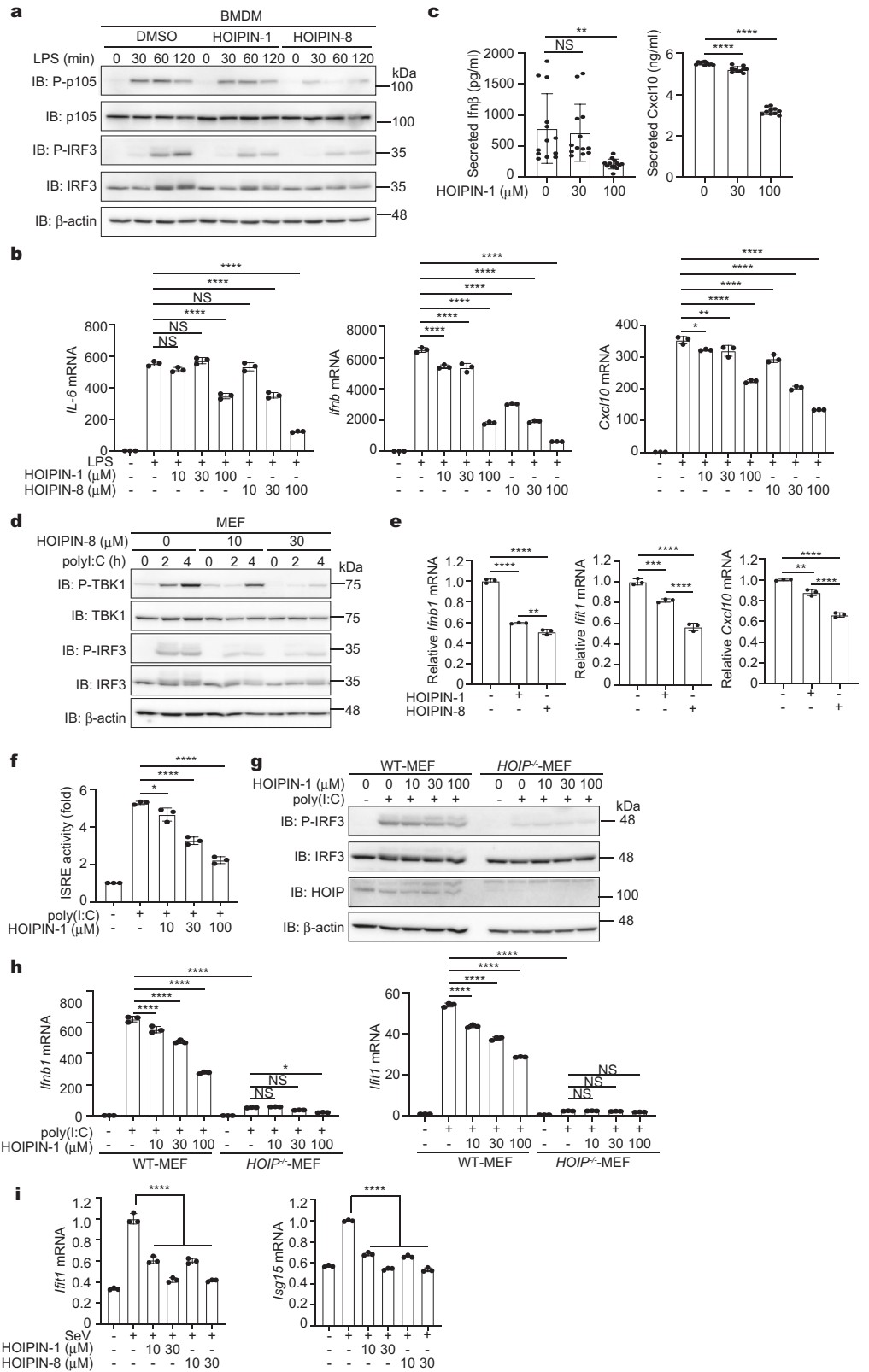

HOIPINs (Fig. 3d). Therefore, HOIPINs appear to have selectivity toward LUBAC.

To investigate the inhibitory mechanism of HOIPINs on the LUBAC activity, we performed a mass spectrometric analysis using recombinant petit-LUBAC, which was prepared by the co-expression of HOIP$^{474–1072}$ and HOIL-1L$^{1–191}$ complex. In the presence of HOIPIN-1, the molecular mass of the truncated HOIP

increased by 263 (Fig. 3e). This suggested that HOIPIN-1 stoichiometrically binds to HOIP. To further clarify the biochemical mechanism of the HOIPIN-1-mediated LUBAC inhibition, we examined thioester-linked ubiquitin transfer. The His-ubiquitin–HOIP intermediates were detected in petit-LUBAC, HOIP$^{474–1072}$ fragment alone, and combined addition of HOIP$^{474–1072}$ and HOIL-1L$^{1–191}$ fragments (Fig. 3f, lanes 1, 2,

**Fig. 2 HOIPINs inhibit PAMPs-induced activation of NF-κB and IFN antiviral pathways. a** HOIPINs suppress the LPS-mediated NF-κB and IFN antiviral pathways. BMDM cells were pre-treated with 30 μM HOIPINs for 30 min, and stimulated with 20 μg/ml LPS for the indicated period with HOIPINs. The cell lysates were immunoblotted with the indicated antibodies. **b** LPS-induced gene expression is suppressed by HOIPINs. BMDM cells were pre-treated with the indicated concentrations of HOIPINs for 30 min, and stimulated with 100 ng/ml LPS for 1 h. The mRNA levels were assessed by qPCR. **c** Suppression of IRF3 targets by HOIPIN-1. BMDM cells were stimulated with 100 ng/ml LPS for 8 h with HOIPIN-1, and interferon β ($n = 13$) and Cxcl10 ($n = 10$) were quantified by ELISA. **d** Suppression of antiviral signaling by HOIPIN-8. MEFs were stimulated with 10 μg/ml poly(I:C) for the indicated period with HOIPIN-8, and subjected to immunoblotting analysis. **e** Suppression of IRF3 targets by HOIPINs. BMDM cells were pre-treated with 30 μM HOIPINs for 30 min, and stimulated with 10 μg/ml poly(I:C) for 2 h with HOIPINs. The mRNA levels were assessed by qPCR. **f** HOIPIN-1 inhibits ISRE-luciferase activity. MEF cells, transfected with the ISRE-luciferase reporter, were stimulated with 10 μg/ml poly(I:C) with or without HOIPIN-1 for 6 h, and the luciferase activities were analyzed. **g** LUBAC activity is indispensable for the IFN pathway. WT- and $HOIP^{-/-}$-MEFs were treated with 10 μg/ml poly(I:C) and HOIPIN-1 for 2 h. Cell lysates were subjected to immunoblotting analysis. **h** HOIP is critical for the expression of IRF3-target genes. WT- and $HOIP^{-/-}$-MEFs were treated as in **g**, and qPCR analyses were performed. **i** The Sendai virus (SeV)-induced antiviral response is suppressed by HOIPINs. MEFs were infected with SeV at a multiplicity of infection (MOI) of 10 for 8 h, and treated with the indicated concentrations of HOIPINs for 30 min. qPCR analyses were performed. In **b, c, e, f, h, i**, data are shown as mean ± SEM, $n = 3$ (sample numbers in **c** are indicated in the legend). NS not significant, $*P < 0.05$, $**P < 0.01$, $***P < 0.001$, $****P < 0.0001$, one-way ANOVA with Tukey's post hoc test.

## Table 1 Data collection and refinement statistics (molecular replacement).

|  | HOIP–HOPIN-1 | HOIP–HOPIN-8 |
|---|---|---|
| **Data collection** | | |
| Space group | $P2_12_12_1$ | $C2$ |
| Cell dimensions | | |
| $\quad a, b, c$ (Å) | 39.4, 60.2, 92.3 | 151.6, 88.8, 104.4 |
| $\quad \alpha, \beta, \gamma$ (°) | 90, 90, 90 | 90, 101.1, 90 |
| Resolution (Å) | 50–1.54 (1.64–1.54) | 50–2.12 (2.25–2.12) |
| $R_{merge}$ | 0.062 (1.580) | 0.117 (1.783) |
| $I/\sigma I$ | 18.22 (1.10) | 9.45 (1.19) |
| Completeness (%) | 99.6 (97.8) | 99.3 (98.6) |
| Redundancy | 8.68 (6.39) | 7.04 (7.20) |
| **Refinement** | | |
| Resolution (Å) | 46.1–1.54 | 46.59–2.12 |
| No. of reflections | 32,894 | 76,185 |
| $R_{work}/R_{free}$ | 19.8/21.7 | 21.1/26.7 |
| No. of atoms | | |
| $\quad$ Protein | 1461 | 9385 |
| $\quad$ Ligand/ion | 65 | 285 |
| $\quad$ Water | 127 | 269 |
| B-factors | | |
| $\quad$ Protein | 38.2 | 60.0 |
| $\quad$ Ligand/ion | 46.0 | 61.6 |
| $\quad$ Water | 44.5 | 55.2 |
| R.m.s. deviations | | |
| $\quad$ Bond lengths (Å) | 0.008 | 0.010 |
| $\quad$ Bond angles (°) | 1.445 | 1.183 |

Values in parentheses are for highest-resolution shell.

and 4), and HOIPIN-1 inhibited the formation of the intermediates (Fig. 3f, lanes 5, 6, and 8). In contrast, HOIPIN-1 did not inhibit the thioester-linked His-ubiquitin–UbcH5C intermediate. Moreover, the formation of the His-ubiquitin–HOIP intermediate was inhibited under the reducing conditions in the absence of HOIPIN-1, suggesting mostly thioester binding (Supplementary Fig. 7d). In contrast, linear polyubiquitination by wild-type ubiquitin remained unchanged by the reducing agent, and these modifications were completely suppressed in the presence of HOIPIN-1. Collectively, these results indicated that HOIPIN-1 selectively suppresses the RING-HECT-hybrid reaction of LUBAC, as represented by the thioester-linked donor ubiquitin transfer from E2 to the active Cys885 of HOIP.

## Crystal structures of HOIPINs-bound HOIP.

Crystal structures of HOIPINs-bound HOIP. To understand the molecular mechanism of HOIP inhibition by HOIPINs, the crystal structures of the RING2-LDD domain of HOIP[10] in the complexes with HOIPIN-1 and HOIPIN-8 were determined at 1.54 and 2.19 Å resolutions, respectively (Table 1). HOIPINs are covalently attached to the catalytic Cys885 of HOIP via Michael addition (Fig. 4a). The benzoate moiety of HOIPINs is the key element for binding to HOIP (Fig. 4b, c, Supplementary Fig. 8a, b). The carboxyl group of the benzoate hydrogen bonds with Arg935, and the aromatic ring of the benzoate hydrophobically interacts with Phe905, Leu922, and the aliphatic portion of His887. The aromatic ring of the methoxyphenyl moiety of HOIPIN-1 and the 2,6-difluorophenyl moiety of HOIPIN-8 stack on the carboxyl group of the benzoate moiety and the guanidino group of Arg935, respectively. Furthermore, the two pyrazol moieties of HOIPIN-8 form additional interactions with HOIP that enhance the affinity (Fig. 4c, Supplementary Fig. 8b). Thus, the 1H-pyrazol-4-yl at the 4-position of the 2,6-difluorophenyl moiety forms a hydrogen bond with Asp936, and the 1-methyl-1H-pyrazol-4-yl at the 4-position of the benzoate moiety is accommodated in the hydrophobic pocket formed by Phe905 and Leu922, and stacks on His887.

We have developed eight types of HOIP inhibitors (HOIPIN-1–HOIPIN-8)[29]. The HOIPIN-1- and HOIPIN-8-bound structures of HOIP explain the differences in the inhibition activities of the eight HOIP inhibitors with petit-LUBAC (Supplementary Fig. 8c). The introduction of a 1-methyl-1H-pyrazol-4-yl group to the benzoate moiety increased the petit-LUBAC inhibition activity 3.7- to 6.1-fold (HOIPIN-2, -6, and -8 were compared to HOIPIN-1, -5, and -7, respectively). It is likely that the 1-methyl-1H-pyrazol-4-yl group of HOIPIN-2 and -6 interacts with HOIP in a manner similar to that of HOIPIN-8 (Fig. 4c, Supplementary Fig. 8b). The effects of replacing the fluorine at the 4-position of the 2,4,6-difluorophenyl group of HOIPIN-3 with the 3-pyridinyl (HOIPIN-4), 4-amino-3-pyridinyl (HOIPIN-5), or 1H-pyrazol-4-yl group (HOPIN-7) have also been analyzed[29]. The inhibitory activity of HOIPIN-7 is 10-fold higher than that of HOIPIN-3, in agreement with the finding that the NH group of the 1H-pyrazol-4-yl moiety of HOIPIN-8 hydrogen bonds with Asp936 (Fig. 4c and Supplementary Fig. 8c). On the other hand, the pyridinyl moieties of HOIPIN-4 and HOIPIN-5 lack the NH group, and thus the inhibitory activities of HOIPIN-4 and HOIPIN-5 are 11- and 3.9-fold lower than that of HOIPIN-7, respectively.

The sequence alignment of the RBR E3 ligases indicates that the LDD domain is unique to HOIP (Fig. 4d). The benzoate, 2,6-difluorophenyl, and 1H-pyrazol-4-yl moieties of HOIPIN-8 interact with Arg935 and Asp936, which are located in the LDD domain of HOIP. Arg935 and Asp936 in the LDD domain reportedly interact with Glu16 and Thr14 in the acceptor

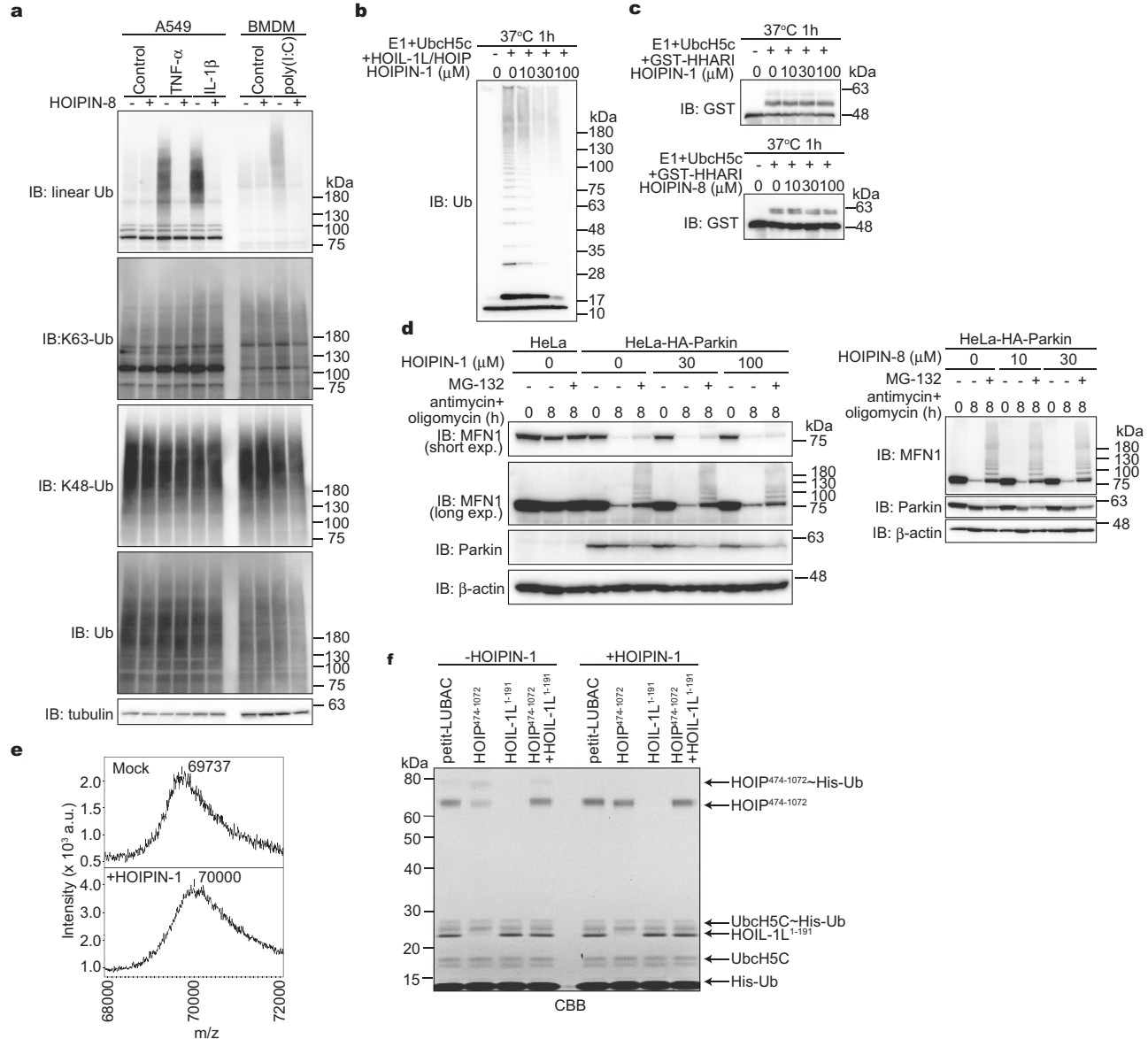

**Fig. 3 HOIPINs suppress the RING-HECT-hybrid reaction of LUBAC. a** HOIPIN-8 selectively suppresses the linear ubiquitin level. A549 cells and BMDM were pre-treated with 30 μM HOIPIN-8, and either 10 ng/ml TNF-α, 1 ng/ml IL-1β, or 10 μg/ml poly(I:C) for 1 h with HOIPIN-8, and cell lysates were immunoblotted with the depicted antibodies. **b**, **c** Effects of HOIPINs on RBR-type E3 activities. In vitro ubiquitination assays for baculovirus-expressed recombinant LUBAC, composed of full-length HOIL-1L and HOIP (**b**), and auto-ubiquitination activities of GST-HHARI (**c**) were examined in the presence of the indicated concentrations of HOIPIN-1 or HOIPIN-8, and immunoblotted with the indicated antibodies. **d** HOIPIN-8 does not suppress the parkin-dependent ubiquitination of mitofusin-1 (MFN1). Parental HeLa and stable HA-parkin-expressing HeLa cells were pre-treated with the indicated concentrations of HOIPINs, and then treated with 5 μM antimycin and 10 μM oligomycin for the indicated times, in the presence of HOIPINs, with or without 10 μM MG-132. The cell lysates were then immunoblotted with the indicated antibodies. **e** HOIPIN-1 stoichiometrically binds to the C-terminal portion of HOIP. Petit-LUBAC was treated with 100 μM HOIPIN-1 for 1 h, and the molecular mass of HOIPIN-1-treated petit-LUBAC was analyzed by mass spectrometry. **f** HOIPIN-1 inhibits the RING-HECT-hybrid reaction in HOIP. In vitro ubiquitination and thioester-linked ubiquitin-binding assay was performed using E1, UbcH5C, His-ubiquitin, and LUBAC components in the presence or absence of 30 μM HOIPIN-1 as indicated. Samples were electrophoresed under non-reducing conditions and stained with Coomassie Brilliant Blue (CBB).

ubiquitin, and the Ala mutations of these residues abrogated the linear ubiquitination activity[9–12]. Thus, HOIPIN-8 not only interacts with the active Cys885 but also masks the critical residues for acceptor ubiquitin-binding. In addition, the 1-methyl-1H-pyrazol-4-yl moiety of HOIPIN-8 interacts with His887, Phe905, and Leu922 of HOIP. His887 and Phe905 are located in the RING2 domain and are partially conserved in Parkin and HHARI (Fig. 4d, e). We constructed 11 HOIP mutants in the HOIPINs-binding sites, and examined the NF-κB

luciferase reporter activity upon the co-expression with HOIL-1L and SHARPIN (Supplementary Fig. 8d). Most of the HOIP mutants failed to induce NF-κB activation, but the F905A mutant showed substantial (~16% of wild type) NF-κB activity. Importantly, the NF-κB activity induced by the F905A mutant of HOIP was insensitive to HOIPIN-8, although the WT-HOIP-mediated NF-κB activation was strongly suppressed by HOIPIN-8 (Fig. 4f). Phe905 of HOIP is replaced by a functionally equivalent Trp residue in Parkin and HHARI. Therefore, the

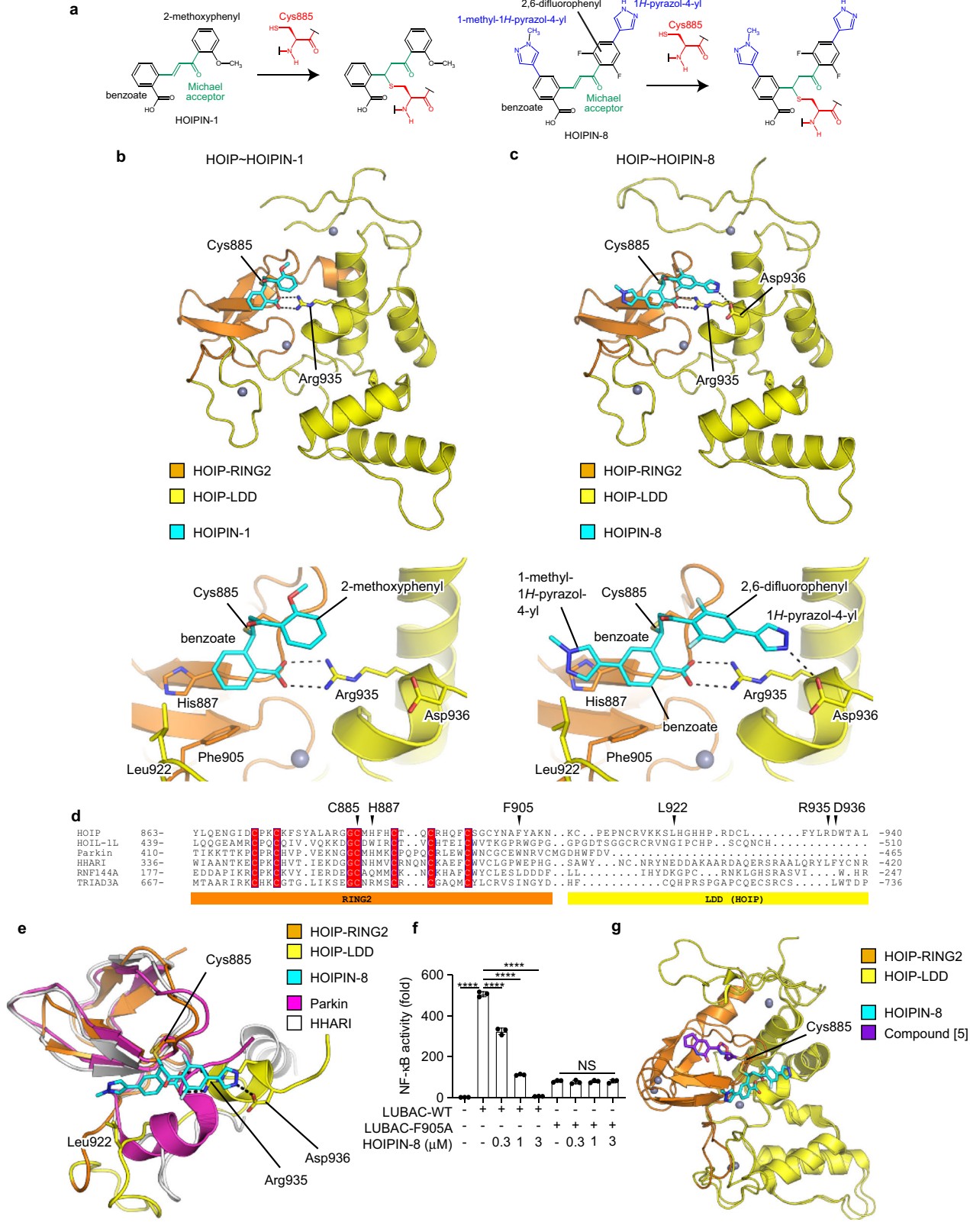

1-methyl-1*H*-pyrazol-4-yl moiety is likely to contribute to the affinity, rather than the specificity for HOIP.

Recently, Rittinger's group developed LUBAC inhibitors and determined the crystal structure of HOIP in complex with one of the inhibitors (designated as compound [5])[27]. Compound [5] was covalently attached to the catalytic Cys885 of HOIP via Michael addition (Supplementary Fig. 8e), although the interaction sites of HOIPINs and compound [5] do not overlap with each other. Compound [5] is accommodated in a hydrophobic pocket formed by Tyr878, Leu880, and Phe888, and hydrogen bonds with the main-chain CO and NH groups of His889 and the Oγ atom of Ser899 (Fig. 4g, Supplementary Fig. 8f). These compound [5]-interacting residues of HOIP are located in the RING2 domain, but are not conserved in other

**Fig. 4 Structural basis for the selective inhibitory effects of HOIPINs on LUBAC. a** Scheme for the Michael addition of HOIPIN-1 or HOIPIN-8 onto the active Cys885 in HOIP. The catalytic Cys885 and the Michael acceptor are shown in red and green, respectively. The additional aromatic rings of HOIPIN-8 are shown in blue. **b, c** Structures of the HOIPIN-1- and HOIPIN-8-bound HOIP complexes. Crystal structures of the HOIP RING2-LDD domain complexed with HOIPIN-1 (**b**) or HOIPIN-8 (**c**) are indicated. Hydrogen bonds are indicated by dashed lines, and crucial residues for interactions with HOIPINs are shown. The upper panels are overall views. The lower panels are close-up views of the HOIPIN-binding sites. **d** Multiple amino acid sequence alignment of RING2 and the following region of the RBR family E3s. Conserved residues in RING2 are highlighted, and the residues interacting with HOIPINs are indicated by arrowheads. **e** Structural bases for the selectivity of HOIPIN-8 to HOIP. The structure of the HOIPIN-8-bound HOIP RING2-LDD is compared with the comparable regions of parkin (PDB 4BM9) and HHARI (PDB 5UDH). **f** The F905A mutant of HOIP is insensitive to HOIPIN-8. NF-κB luciferase activities induced by wild-type HOIP (LUBAC-WT) or the F905A mutant (LUBAC-F905A) with HOIL-1L and SHARPIN were assessed in the presence of various concentrations of HOIPIN-8. Data are shown as mean ± SEM, $n = 3$. NS not significant, $****P < 0.0001$, one-way ANOVA with Tukey's post hoc test. **g** Different interactions of HOIPIN-8 and compound [5] with HOIP. Structural comparisons of HOIPIN-8- or compound [5]-bound HOIP (PDB 6GZY) are indicated.

RBR E3s (Fig. 4d). These sequence variations of RING2 may be beneficial for the HOIP specificity of compound [5]. In contrast, the LDD domain of HOIP may be a key determinant for the HOIP specificity of HOIPINs. Therefore, the mechanism underlying the HOIP specificity is completely different between HOIPINs and compound [5].

**HOIPINs accelerate TNF-α-induced extrinsic apoptosis.** TNF-α-mediated NF-κB activation basically functions as anti-apoptotic system; however, TNF-α causes apoptosis when NF-κB-targeted anti-apoptotic gene expression is abolished by a protein synthesis inhibitor, such as cycloheximide (CHX)[39]. A deficiency of LUBAC subunits enhances TNF-α-mediated apoptosis[14,40–42], suggesting that the LUBAC activity is crucial for anti-apoptosis. We first confirmed that treatments with HOIPINs to 100 μM showed no apparent cytotoxicity in A549 cells (Fig. 5a, Supplementary Fig. 9a, Supplementary Table 1). In the presence of TNF-α, but not IL-1β, HOIPIN-8 slightly accelerated cell death (Supplementary Fig. 9b, c), and the combined addition of TNF-α + CHX with HOIPINs decreased the cell viability and accelerated cell death (Fig. 5b, c, Supplementary Fig. 9d, e). The combined treatment of HOIPIN-1 with TNF-α + CHX enhanced the expression of apoptotic factor mRNAs, such as *BAX* and *PUMA* (Supplementary Fig. 9f). Furthermore, HOIPINs increased the TNF-α + CHX-induced cleavage of caspases and PARP (Fig. 5d, Supplementary Fig. 9g). The enhanced TNF-α-mediated cell death by HOIPIN-1 was suppressed by a caspase inhibitor, ZVAD (Fig. 5e), and the formation of the pro-apoptotic TNFR complex II, composed of caspase 8, RIP1, and FADD[43], was also enhanced in the presence of HOIPIN-1 (Fig. 5f). Thus, HOIPINs enhance TNF-α-mediated apoptosis.

The spontaneous deficiency of the *Sharpin* in mice (*cpdm* mice) causes enhanced apoptosis and severe dermatitis[15,19,40]. Indeed, *cpdm* MEF cells showed higher contents of trypan blue-positive cells than those in A549 and wild-type (WT) MEF under basal conditions (Supplementary Fig. 9h). In *cpdm* MEF cells, a treatment with HOIPIN-1 alone showed no effect, whereas the combined addition with TNF-α or TNF-α + CHX enhanced cell death as compared to WT-MEF cells (Supplementary Fig. 9h, Supplementary Table 1). In contrast, HOIPIN-1 had no effects on cell death induced by genotoxic agents (Supplementary Fig. 9i).

To further investigate the effect of HOIPIN-8 on cell death, we constructed *HOIP*-deficient HeLa cells and the C885A mutant of HOIP-restored cells (Supplementary Fig. 10a). Although the cell viability in the parental HeLa cells was not affected by TNF-α + CHX, it was reduced in the *HOIP*-KO and HOIP-C885A-expressing cells (Supplementary Fig. 10b). Moreover, HOIPIN-8 accelerated cell death in the parental HeLa cells, but not in the *HOIP*-KO and HOIP-C885A-expressing cells. Since ZVAD completely suppressed cell death in these cells, apoptosis is enhanced in HeLa cells, which do not express RIP3. Thus, the

HOIPIN-8-mediated apoptosis is highly dependent on LUBAC. To investigate the involvement of necroptosis, we examined TNF-α alone-induced cell death in RIP3-expressing MEF and Jurkat cells (Supplementary Fig. 10c, d). In WT-MEFs, the cell death induced by TNF-α, HOIPIN-8, and ZVAD was completely suppressed by necrostatin-1, suggesting that the TNF-α-mediated necroptosis is specifically induced with the inhibition of LUBAC and apoptosis. In contrast, in *cpdm* MEFs, TNF-α-mediated necroptosis was induced in the absence of HOIPIN-8, although the co-treatment with HOIPIN-8 and ZVAD further enhanced the cell death (Supplementary Fig. 10c). In the parental Jurkat cells, the combined treatment with TNF-α and HOIPIN-8 induced cell death. Since both ZVAD and necrostatin-1 showed partial suppressive effects, apoptosis and necroptosis were simultaneously induced in Jurkat cells (Supplementary Fig. 10d). Intriguingly, *HOIP*-deficient Jurkat cells were labile for TNF-α-induced cell death, and the co-treatment with TNF-α and HOIPIN-8 caused severe cell death, which was partially rescued by ZVAD and necrostatin-1. Collectively, these results indicate that the LUBAC activity is indispensable for the TNF-α-induced extrinsic apoptotic and necroptotic pathways.

**HOIPINs induce cell death in B cell lymphoma cell lines.** In diffuse large B cell lymphoma (DLBCL), activated B cell-like DLBCL (ABC-DLBCL) is reportedly induced by constitutively elevated NF-κB activation, whereas the germinal center B cell-like DLBCL (GCB-DLBCL) is associated with defects in chromatin remodeling[44,45]. Importantly, the polymorphisms in *HOIP* are closely associated with ABC-DLBCL[23]. Moreover, the *HOIP* knockdown reportedly reduced the viability of ABC-DLBCL cells[46]. Therefore, we investigated the effect of HOIPINs on B cell lymphoma cells. We confirmed that the viability of ABC-DLBCL cell lines, but not that of GCB-DLBCL cell lines, was remarkably suppressed in the presence of HOIPIN-1 (Fig. 6a, Supplementary Fig. 11a). Indeed, HOIPIN-1 showed lower $IC_{50}$ values with the ABC-DLBCL cell lines than with the GCB-DLBCL cell lines, and HOIPIN-8 showed more potent inhibitory effects than those of HOIPIN-1 (Supplementary Table 1, Fig. 6b, Supplementary Fig. 11b). In ABC-DLBCL cell lines, the cleavages of caspase 3 and PARP were enhanced in the presence of HOIPIN-8 (Fig. 6c), and the cell death was suppressed by ZVAD (Supplementary Fig. 11c), suggesting that HOIPINs effectively induce apoptosis in ABC-DLBCL cells. In ABC-DLBCL cell lines, the phosphorylation of p105 and p65 and the expression of NF-κB target genes were suppressed by HOIPINs (Fig. 6d, e). Moreover, the phosphorylation of IκBα was decreased concomitantly with the reduced intracellular amounts of linear polyubiquitin in HOIPIN-1-treated HBL1 cells (an ABC-DLBCL cell line), but not in BJAB cells (a GCB-DLBCL cell line) (Fig. 6f). HOIPIN-8 more effectively suppressed linear ubiquitination-mediated NF-κB activation and gene expression, as compared to HOIPIN-1 (Fig. 6g,

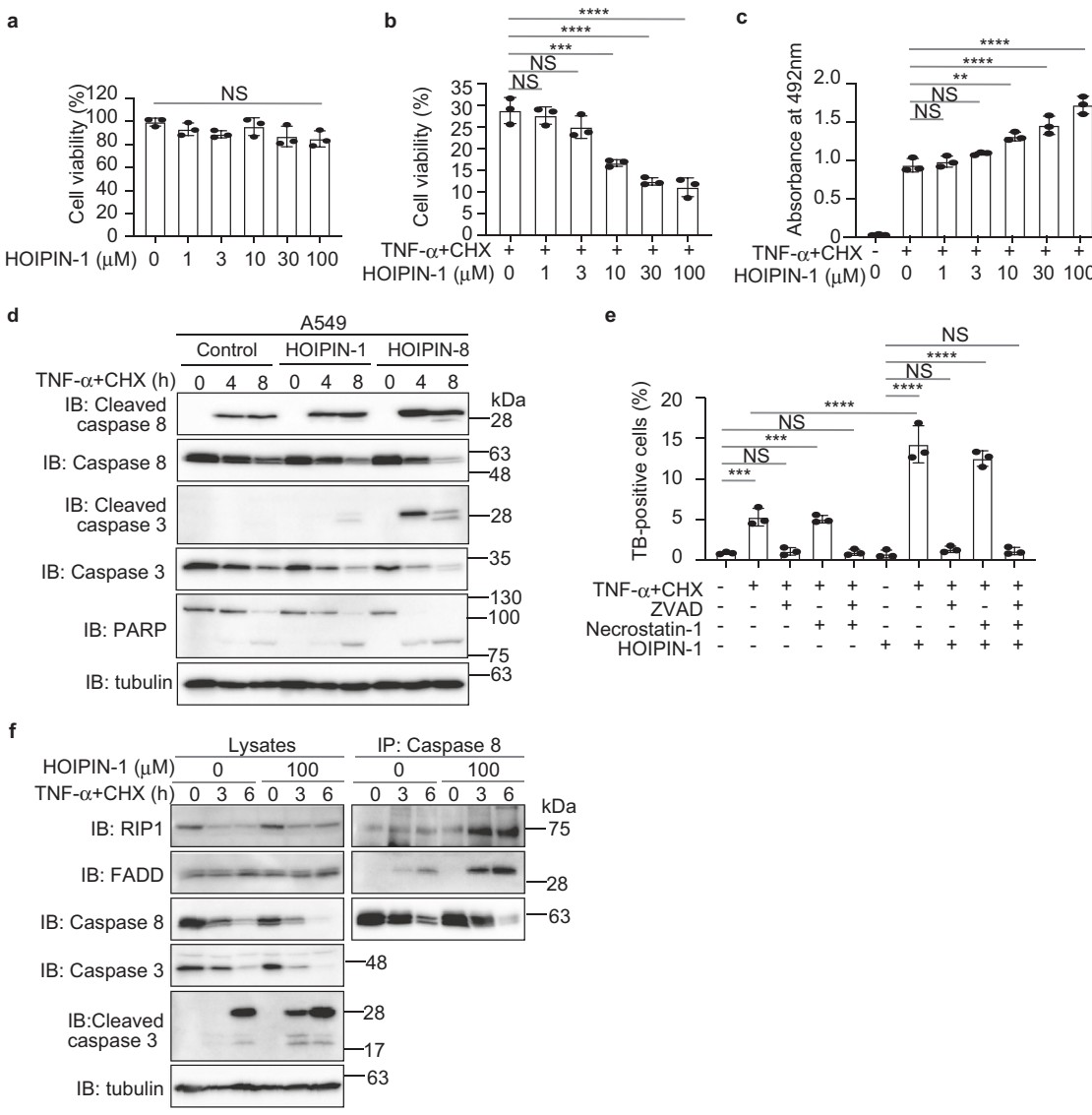

**Fig. 5 HOIPINs accelerate TNF-α-induced apoptosis. a** HOIPIN-1 alone shows no cytotoxicity. A549 cells were treated with the indicated concentrations of HOIPIN-1 for 48 h, and the cell viability was assayed by Calcein-AM. **b** HOIPIN-1 decreases the viability of TNF-α-treated cells. A549 cells were pretreated with the indicated concentrations of HOIPIN-1 for 1 h. The cells were then treated with 40 ng/ml TNF-α and 20 μg/ml CHX in the presence of HOIPIN-1 for 48 h. The cell viability was assayed by Calcein-AM, as in **a**. **c** HOIPIN-1 accelerates TNF-α-induced cell death. A549 cells were treated as in **b**, and the cell toxicity was analyzed by the lactate dehydrogenase activity. **d** Caspase activation in HOIPINs-treated cells. A549 cells were pre-treated with 10 μM HOIPIN-1 or HOIPIN-8 for 1 h. The cells were then treated with 5 ng/ml TNF-α + 5 μg/ml CHX in the presence of HOIPIN-1 or HOIPIN-8, and the cell lysates were immunoblotted with the indicated antibodies. **e** HOIPINs induce TNF-α-mediated apoptosis. A549 cells were pre-treated with 100 μM HOIPIN-1 for 1 h. The cells were then treated with 40 ng/ml TNF-α + 20 μg/ml CHX, 100 μM HOIPIN-1, 20 μM ZVAD, and/or 100 μM necrostatin-1 for 14 h, as indicated, and trypan blue-positive cells were counted. **f** Enhanced TNF receptor complex II formation in HOIPIN-1-treated cells. A549 cells were pre-treated with 100 μM HOIPIN-1 for 30 min. The cells were then treated with 40 ng/ml TNF-α + 20 μg/ml CHX, in the presence or absence of 100 μM HOIPIN-1, for the indicated periods. Cell lysates were immunoprecipitated with an anti-caspase 8 antibody, and immunoblotted with the indicated antibodies. In **a**, **b**, **c**, **e**, data are shown as mean ± SEM, $n = 3$, NS not significant, **$P < 0.01$, ***$P < 0.001$, ****$P < 0.0001$, by one-way ANOVA with Tukey's post hoc test.

Supplementary Fig. 11d). These results indicate that the inhibition of LUBAC by HOIPINs suppressed the linear ubiquitination-associated NF-κB activation in ABC-DLBCL cells, leading to apoptosis.

**HOIPIN-1 alleviates psoriasis**. Psoriasis vulgaris is a chronic inflammatory- and immune-mediated skin disease[47]. To investigate the therapeutic effects of HOIPIN-1 in vivo, we treated imiquimod-induced psoriasis-model mice with HOIPIN-1. The administration of imiquimod (IMQ) cream on the shaved back

skin of BALB/c mice for 5 days generated psoriasis-like phenotypes, as described[48]. Although the treatment with HOIPIN-1 alone showed no apparent effect, the treatment with HOIPIN-1 and IMQ as well as the treatment with clobetasol propionate, a corticosteroid used for psoriasis therapy[49], suppressed these symptoms (Fig. 7a). We evaluated and scored the erythema, thickness, and scaling on a scale from 0 to 4[48], and the cumulative score suggested that HOIPIN-1 suppressed IMQ-induced psoriasis after 4 days of treatment (Fig. 7b). Analyses of the H&E-stained sections indicated that the imiquimod-induced increase of epidermal thickening was suppressed by clobetasol and

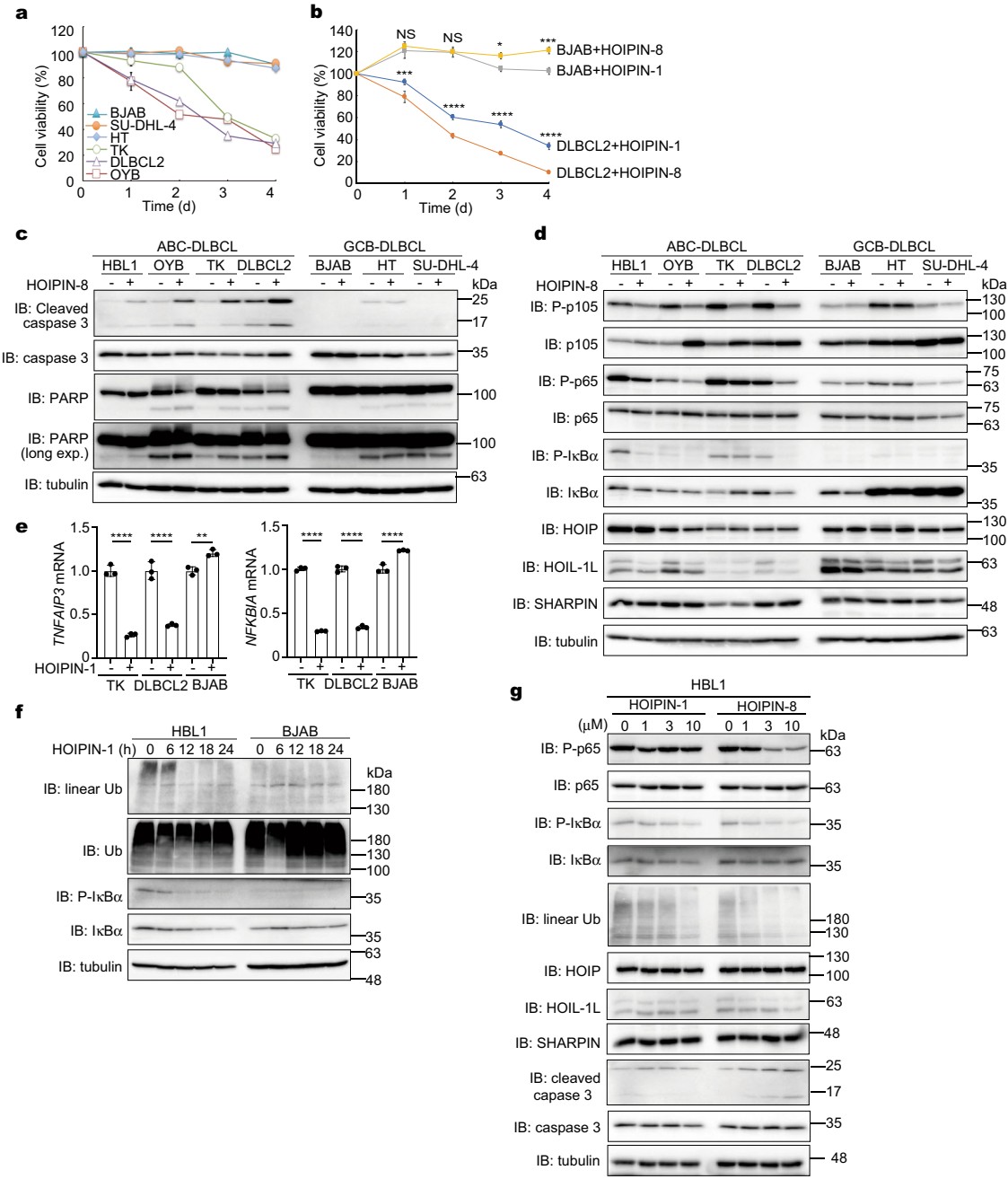

**Fig. 6 HOIPINs efficiently suppress ABC-DLBCL cell lines. a** HOIPIN-1 shows potent toxicity to ABC-DLBCL cell lines. GCB-DLBCL cells (BJAB, SU-DHL-4, and HT) and ABC-DLBCL cells (TK, DLBCL2, and OYB) were cultured in the presence of DMSO or 10 μM HOIPIN-1. Taking the cell viabilities in the presence of DMSO as 100%, the relative cell viabilities of respective cell line in the presence of HOIPIN-1 were assessed by a CellTiter-Glo luminescent cell viability assay. **b** HOIPIN-8 enhanced the cell death of ABC-DLBCL cells than HOIPIN-1. DLBCL2 cells or BJAB cells were treated with DMSO, 10 μM HOIPIN-1, or 10 μM HOIPIN-8 for the indicated period, and the relative cell viabilities were assessed as in **a**. **c** Caspase activation in ABC-DLBCL cells by HOIPIN-8. Cells were treated with or without 10 μM HOIPIN-8 for 24 h, and cell lysates were immunoblotted with the indicated antibodies. **d** Reduced NF-κB activation in ABC-DLBCL cells by HOIPIN-8. Cells were treated and analyzed as in **c**. **e** HOIPIN-1 suppresses the expression of NF-κB target genes in ABC-DLBCL cells. Cells were treated with 10 μM HOIPIN-1 for 24 h, and qPCR analyses were performed. **f** Intracellular linear ubiquitin and IκBα phosphorylation in ABC-DLBCL cells are diminished by HOIPIN-1. HBL1 and BJAB cells were treated with 10 μM HOIPIN-1 for the indicated period, and cell lysates were immunoblotted with the indicated antibodies. **g** HOIPIN-8 has potent inhibitory effects on NF-κB activation and linear ubiquitination in ABC-DLBCL cells than those in HOIPIN-1. HBL1 cells were treated with the indicated concentrations of HOIPIN-1 or -8 for 24 h, and cell lysates were immunoblotted with the indicated antibodies. In **b**, **e**, data are shown as mean ± SEM, NS not significant, *$P < 0.05$, **$P < 0.01$, ***$P < 0.001$, ****$P < 0.0001$, by one-way ANOVA with Tukey's post hoc test.

HOIPIN-1 after 7 days (Fig. 7c, d), suggesting that HOIPIN-1 has therapeutic effects on the hyperproliferation of keratinocytes. Although the clobetasol monotherapy induced epidermal thinning and reduced the Ki-67-positive cells, which are a marker of

cell proliferation in the epidermis, these side effects were not observed in the HOIPIN-1-treated mice (Fig. 7c, Supplementary Fig. 12a). Moreover, the IMQ-treatment did not induce TUNEL-positive apoptotic cells, and the administration of HOIPIN-1 had

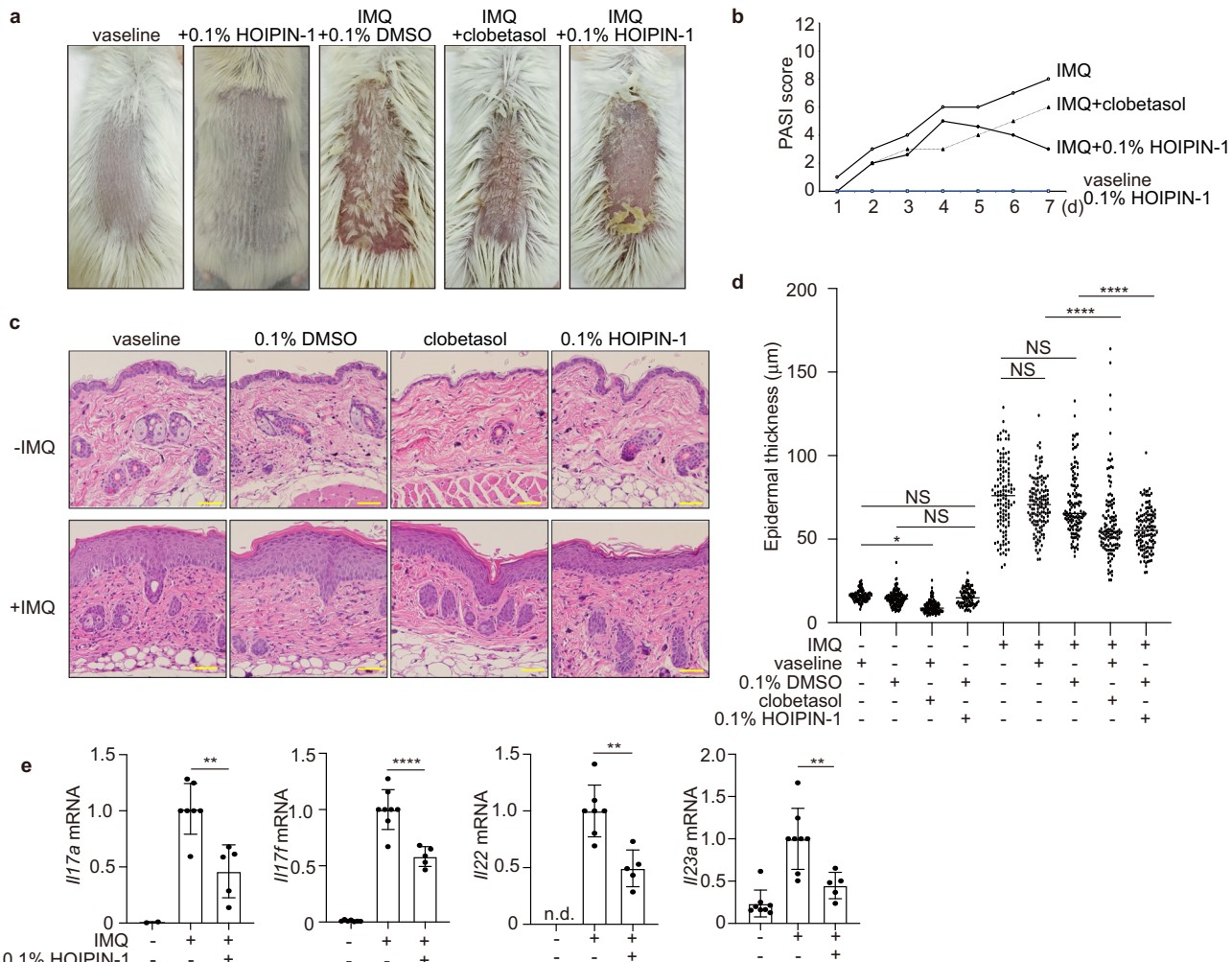

**Fig. 7 HOIPIN-1 shows therapeutic effects on psoriasis. a** Phenotypical presentation of mouse back skin after 5 days of treatment. BALB/c mice (8-week-old female) were treated daily with control vaseline or imiquimod cream, in the presence of clobetasol or 0.1% HOIPIN-1 in DMSO. **b** HOIPIN-1 alleviates imiquimod-induced psoriasis. Erythema, scaling, and thickness of the back skin were scored daily on a scale from 0 to 4, and the cumulative scores (means, $n = 3$) are shown. **c** HOIPIN-1 reduces the thickened epidermis induced by the imiquimod treatment. Mice were treated for 7 days with imiquimod, clobetasol, and/or HOIPIN-1, as indicated. H&E staining of the back skin was performed. Bars, 50 μm. **d** The thickness of the epidermis, at 90–132 sites from three to four mice treated as indicated, was measured using the ImageJ software (National Institutes of Health, Bethesda, MD) and statistically analyzed. **e** HOIPIN-1 suppresses cytokine expression in the back skin of imiquimod-treated mice. The mRNA levels of *Il17a*, *Il17f*, *Il22*, and *Il23a* in the back skin of the mice were examined by TaqMan PCR. Data are shown as means ± SEM. In **d**, **e**, NS not significant, *$P < 0.05$, **$P < 0.01$, ****$P < 0.0001$, by one-way ANOVA with Tukey's post hoc test.

no effect on the contents of TUNEL-positive cells (Supplementary Fig. 12b–d). Importantly, the mRNA levels of inflammatory cytokines, such as *Il17*, *Il22*, and *Il23*, were suppressed by HOIPIN-1 (Fig. 7e). These cytokines are crucial to generate Th1, Th17, and group 3 innate lymphoid cells (ILC3) for the progression of psoriasis[47,50]. Therefore, these results indicate that HOIPIN-1 shows anti-inflammatory effects on T cells, but does not affect cell death, in vivo in psoriasis-model mice.

## Discussion

The LUBAC-mediated linear ubiquitination activity plays important roles in the regulation of innate immune responses[8,51]. The modulation of the LUBAC activity through its inhibitors would be useful to study innate immunity and therapeutic drug seeds. Among reported LUBAC inhibitors, BAY11-7082 inhibits E2s; therefore, it may not be a LUBAC-directed inhibitor[21]. Gliotoxin binds HOIP and suppresses the in vitro LUBAC activity, with an IC$_{50}$ value of 0.51 μM, but cytotoxic[22]. Although

bendamustine suppresses the in vitro LUBAC activity (IC$_{50}$ = 6.3 μM), it also substantially suppresses other E3s, such as ITCH and MDM2 (ref. [26]). Moreover, an α,β-unsaturated methyl ester-containing compound, compound [11a], inhibits the over-expressed LUBAC-induced NF-κB activity, with an IC$_{50}$ value of 37 μM in HEK293T cells[27]. Unexpectedly, however, BAY11-7082, gliotoxin, and bendamustine had little suppressive effects on NF-κB activation and linear ubiquitin generation at non-toxic concentrations in LUBAC-expressing cells (Supplementary Fig. 1). We described HOIPIN-1 (ref. [28]) and HOIPIN-8 as inhibitors of LUBAC[29] (Fig. 1a), and that HOIPIN-1 and HOIPIN-8 inhibit the in vitro linear ubiquitination activity of the petit-LUBAC with IC$_{50}$ values of 2.8 μM and 11 nM, respectively. Furthermore, HOIPIN-1 and HOIPIN-8 suppressed the overexpressed LUBAC-induced NF-κB activity in HEK293T cells with IC$_{50}$ values of 4.0 and 0.42 μM[29], respectively, indicating that HOIPIN-8 is the most potent LUBAC inhibitor among the reported LUBAC inhibitors. We determined that HOIPINs are conjugated to the active site Cys885 in the RING2 domain of

HOIP (Figs. 3 and 4, Supplementary Fig. 8), and interrupt the RING-HECT-hybrid reaction. Although the α,β-unsaturated carbonyl-containing chemicals seem to react with various SH-groups through the Michael reaction, HOIPINs did not inhibit the E1-mediated ubiquitin transfer to E2, or the activities of the HECT-, RING-, and other RBR-type E3s, and specifically suppressed the intracellular linear ubiquitin level induced by TNF-α, IL-1β, and poly(I:C) (Fig. 3, Supplementary Fig. 7). Although K63-linked ubiquitin is involved in the NF-κB activation, an increase in the intracellular amounts of this linkage was not detected under our conditions. Since K63-linked ubiquitin is the second-most abundant ubiquitin linkage and functions in a variety of cellular functions, such as DNA repair and membrane trafficking[4], it may not be affected by the stimulation. Otherwise, the K63-linked ubiquitin level may rapidly return to the steady state. These results suggested that HOIPINs show selectivity toward LUBAC-mediated linear ubiquitination in cells, although we cannot deny the possibility that they may react with other off-targets. Indeed, a structural analysis clearly indicated that not only the RING2 domain but also the unique residues in the LDD domain, such as Arg935 and Asp936, contribute to the binding-specificity of HOIPINs to LUBAC (Fig. 4, Supplementary Fig. 8). Since the binding pockets for HOIPINs and compound [5] are totally different[27], the structural information about these chemicals may promote the further improvement of LUBAC inhibitors.

At the cellular level, HOIPINs down-regulate the canonical NF-κB activation pathways mediated by inflammatory cytokines, CD40, T cell receptor, CpG, and LPS (Fig. 1, Supplementary Figs. 2–4). However, HOIPINs had no inhibitory effects on the B cell receptor- and LT-β-mediated NF-κB activation and MAPK pathways, in which LUBAC is uninvolved[7,8,18,19]. Furthermore, HOIPINs showed no inhibitory effects in HOIP-deficient cells. Collectively, these results suggested that LUBAC is the major target of HOIPINs in the inhibition of the canonical NF-κB pathway.

Although the effects of LUBAC on the IFN antiviral pathway have been controversial, several recent reports suggested that the LUBAC activity is crucial for the IRF3 activation[34–36]. Moreover, OTULIN, a linear ubiquitin-specific deubiquitinase, was revealed to down-regulate the IFN induction pathway[52], suggesting that linear ubiquitination seems to regulate antiviral pathway. In this study, we showed that the LPS-, poly(I:C)-, and SeV-mediated IFN production pathway was impaired in HOIP-deficient cells, and HOIPINs suppressed the antiviral pathway through the reduced activation of TBK1 and IRF3 (Fig. 2, Supplementary Figs. 5 and 6). Recently, LUBAC is reportedly indispensable for the TNF-induced TBK1 and IKKε activation and prevention of cell death[53]. Thus, LUBAC and linear ubiquitination play a part in the antiviral pathway.

A deficiency of LUBAC subunits enhances TNF-α-mediated cell death[14,40–42]. We showed that the HOIPINs accelerate the TNF-α-induced extrinsic apoptotic and necroptotic pathways in various cells (Fig. 5, Supplementary Figs. 9 and 10). Furthermore, HOIPINs exhibited potent inhibitory effects on the viability of ABC-DLBCL cells, by suppressing the linear ubiquitination-mediated NF-κB activation and inducing apoptosis (Supplementary Table 1, Fig. 6, Supplementary Fig. 11). Mutations in the genes involved in the NF-κB pathway induce ABC-DLBCL[44]. Furthermore, polymorphisms in HOIP are a risk factor for ABC-DLBCL[23]. Since ABC-DLBCL is difficult to manage, the suppression of the NF-κB activity by HOIPINs or more improved derivatives may provide options to treat the disease.

As an additional in vivo application, we showed that HOIPIN-1 exhibits therapeutic effects on psoriasis (Fig. 7, Supplementary Fig. 12). Psoriasis vulgaris is a chronic inflammatory- and immune-mediated skin disease. To date, more than 70 genes, involved in skin barrier production, IL-23/IL17-mediated lymphocyte signaling, and the NF-κB pathway, reportedly affect the pathogenesis of psoriasis[47,54]. Thus, the chronic inflammation mediated by aberrant NF-κB activation is a critical target to treat psoriasis. We have shown that HOIPIN-1 serves as an additional anti-inflammatory therapeutic tool to treat psoriasis, since it suppresses the expression of IL-23/IL-17, which is crucial for the Th1, Th17, and ILC3 cell-mediated pathogenesis of psoriasis[50], and has no apparent side effects, unlike clobetasol, a steroid drug. It should be noted that spontaneous mutation of Sharpin in mice[15,19,40], and keratinocyte-specific deletion of Hoip or Hoil-1l (ref. [55]) induce severe dermatitis and cell death. Moreover, epidermal-specific ablation of c-IAPs in mice, a the topical application of Smac mimetics, resulting in the depletion of c-IAPs and the reduced activation of LUBAC, also cause dermatitis[56]. Since these results indicate that the excessive reduction of LUBAC activity causes skin lesions, careful titration of HOIPINs is necessary for the appropriate treatment of psoriasis. Based on the HOIPINs, further studies are necessary to create potent and specific inhibitors of LUBAC for therapeutic purposes.

## Methods

**Reagents and plasmids.** HOIPIN-1 (2-[(1E)-3-(2-methoxyphenyl)-3-oxoprop-1-en-1yl] benzoic acid sodium salt) and HOIPIN-8 (2-{(E)-3-[2,6-difluoro-4-(1H-pyrazol-4-yl)-phenyl]-3-oxo-propenyl}-4-(1-methyl-1H-pyarol-4-yl)-benzoic acid sodium salt) were prepared as described[28,29]. The following reagents and kits were obtained as indicated: recombinant human TNF-α (BioLegend), recombinant human IL-1β (BioLegend), CpG-B DNA (Hycult Biotech), LPS from E. coli O111:B4 (Invivogen), high molecular weight poly(I:C) (Invivogen), antimycin (Sigma-Aldrich), etoposide (Sigma-Aldrich), cycloheximide (Sigma-Aldrich), gliotoxin (Sigma-Aldrich), human CD40 ligand (R&D Systems), doxorubicin (Wako), camptothecin (Wako), BAY11-7082 (Wako), bendamustine (Tokyo Chemical Industry), MG-132 (Peptide Institute), oligomycin (Calbiochem), zVAD-FMK (ZVAD)(ENZO Life Sciences), necrostatin-1 (ENZO Life Sciences), E6AP Ubiquitin Ligase Kit (Boston Biochem, K-230), NE-PER Nuclear and Cytoplasmic Extraction Reagents (Thermo, 78833), Human IL-6 DuoSet ELISA Kit (R&D Systems, DY206-05), IFN beta Mouse ELISA kit (Thermo Fisher Scientific, 424001), Mouse CXCL10/IP-10/CRG-2 ELISA Kit (Novus, NBP1-92665).

The open reading frames of human cDNAs[14,19] were amplified by reverse transcription-PCR. Mutants of these cDNAs were prepared by the QuikChange method, and the entire nucleotide sequences were verified. The cDNAs were ligated to the appropriate epitope sequences and cloned into the pcDNA3.1 (Invitrogen), pGEX-6P-1 (GE Healthcare), and pMAL-c2x (New England Biolabs) vectors.

**Antibodies.** The following antibodies were used for western blot and cell stimulation: parkin (Cell Signaling, 2132; 1:1000 for western blot), P-IκBα (Cell Signaling, 9246; 1:1000), IκBα (Cell Signaling, 4812; 1:1000), P-p105 (Cell Signaling, 4806; 1:1000), p105 (Cell Signaling, 3035; 1:1000), P-p65 (Cell Signaling, 3033; 1:1000), p65 (Cell Signaling, 8242; 1:1000), P-IKKα/β (Cell Signaling, 2697; 1:1000), P-JNK (Cell Signaling, 4668; 1:1000), JNK (Cell Signaling, 9252; 1:1000), p100/p52 (Cell Signaling, 4882; 1:1000), P-TBK1 (Cell Signaling, 5483; 1:1000), TBK1 (Cell Signaling, 3504; 1:1000), P-IRF3 (Cell Signaling, 4947 and 37829; 1:1000), IRF3 (Cell Signaling, 4302; 1:1000), caspase 8 (Cell Signaling, 4790; 1:1000), cleaved caspase 8 (Cell Signaling, 9496; 1:1000), caspase 3 (Cell Signaling, 9662; 1:1000), cleaved caspase 3 (Cell Signaling, 9661; 1:1000), PARP (Cell Signaling, 9542; 1:1000), GST (Cell Signaling, 2622; 1:1000), MBP (Cell Signaling, 2396; 1:1000), Lamin A/C (Cell Signaling, 4777; 1:1000), MFN1 (Cell Signaling, 14739; 1:1000), P-TAK1 (Cell Signaling, 4508; 1:500), TAK1 (Cell Signaling, 5206; 1:1000), HOIL-1L (Santa Cruz Biotech, sc-49718; 1:500), optineurin (Santa Cruz Biotech, sc-166576; 1:1000), ubiquitin (P4D1) (Santa Cruz Biotech, sc-8017; 1:1000), TNFR1 (Santa Cruz Biotech, sc-7895; 1:1000), IKKα/β (Santa Cruz Biotech, sc-7607; 1:1000), NEMO (Santa Cruz Biotech, sc-8330; 1:1000), β-actin (Santa Cruz Biotech, sc-47778; 1:1000), caspase 8 (Santa Cruz Biotech, sc-6136), RIP1 (BD Biosciences, 610458; 1:1000), FADD (BD Biosciences, 610399; 1:1000), CD3 (BD Biosciences, 555337), CD28 (BD Biosciences, 555726), linear ubiquitin (LUB9) (Millipore, MABS451; 1:1000), tubulin (Cedarlane, CLT9002; 1:3000), NEMO (MBL, K0159-3; 1:2000), HOIP (Abcam, ab125189; 1:1000), SHARPIN (Proteintech, 14626-1-AP; 1:1000), HA (Roche, 11867423001; 1:1000), DYKDDDDK (1E6; HRP-conjugate) (Wako, 015-22391; 1:10,000), Ki-67 (Dako, M7240), CD40 (HM40-3) (eBioscience, 16-0402-85), LTβR(3C8) (eBioscience, 16-5671-82), IgM (Southern Biotechnology, 2022-01), LTβR(4H8 WH2) (AdipoGen, AG-20B-0008).

**Cell culture.** A549, HEK293T, HeLa (ATCC), optineurin-deficient HeLa[31], HOIP-deficient HeLa, C885A mutant of HOIP-restored HeLa, wild-type MEF, and Sharpin-deficient cpdm MEF cells were cultured in DMEM containing 10% fetal

bovine serum, 100 IU/ml penicillin G, and 100 μg/ml streptomycin. BJAB, Jurkat (ATCC), and *HOIP*-deficient Jurkat[16] cells, ABC-DLBCL cell lines (TK, DLBCL2, OYB, HBL1), and GCB-DLBCL cell lines (SU-DHL-4 and HT), which were kindly provided by Dr. Hitoshi Ohno, were maintained in RPMI containing 10% FBS, 55 μM 2-mercaptoethanol, and antibiotics, at 37 °C under a 5% CO$_2$ atmosphere. To obtain specific cell types, suspensions of mouse splenocytes were prepared from freshly removed mouse spleens, and erythrocytes were depleted with ACK lysing buffer (0.15 M NH$_4$Cl, 10 mM KHCO$_3$, 0.1 mM Na$_2$EDTA, pH 7.2), and then separated by BD IMag anti-mouse CD45R/B220 particles (BD Biosciences) or an IMag Mouse T Lymphocyte Enrichment Set (BD Biosciences). Each fraction was maintained in RPMI containing 10% FBS, 55 μM 2-mercaptoethanol, and anti-biotics, at 37 °C under a 5% CO$_2$ atmosphere. BMDMs were prepared basically as described[57]. Briefly, bone marrow cells were isolated from the femurs and tibias of B6 mice. After the removal of the reticulocytes, the residual cells were cultured in 10% FCS/RPMI, supplemented with 5% of L-929-conditioned medium. The floating cells were removed, and the attached cells were passed every 2 days. The resultant adherent cells were used as macrophages on day 7. Cell cultures were routinely tested for mycoplasma contamination.

**SDS-PAGE and immunoblotting**. Samples were separated by SDS-PAGE and transferred to PVDF membranes. After blocking, the membranes were incubated with the appropriate primary antibodies, and then incubated with HRP-conjugated secondary antibodies. The chemiluminescent images were obtained with a LAS 4000 imaging analyzer (GE Healthcare) or a Fusion Solo S imaging system (Vilber). Uncropped images are showed in Supplementary Figs. 13 and 14.

**Linear ubiquitination assay of endogenous NEMO**. After a treatment with a series of HOIPIN-1 concentrations for 3 h, A549 cells in six-well plates were sti-mulated with 1.5 ng/ml IL-1β for 15 min. The cells were then lysed with 150 μl of lysis buffer, containing 20 mM Tris-HCl (pH 7.5), 1% SDS, 1 mM *N*-ethylmalei-mide, 20 μM MG-132, and complete inhibitor cocktail without EDTA (Roche). The samples were heated at 90 °C for 10 min, and then diluted with 1.35 ml of buffer, containing 20 mM Tris-HCl (pH 7.5) and 1% Triton X-100. The endogenous NEMO was immunoprecipitated with an anti-NEMO antibody (Santa Cruz) and immunoblotted.

**RNA-seq analysis**. A549 cells were pre-treated with dimethyl sulfoxide, 100 μM HOIPIN-1, or 30 μM HOIPIN-8 for 30 min, and stimulated with 1 ng/ml IL-1β for 2 h in the absence or presence of HOIPINs. The cells were then lysed, and the total RNA was extracted using a RNeasy Mini kit (Qiagen, 74104) according to the manufacturer's instructions. The mRNA template library was constructed with a TruSeq Stranded mRNA LT Sample Prep Kit (Illumina), and the sequencing analysis was performed by Macrogen Corp. (Korea), using a NovaSeq 6000 sequencer (Illumina) and an S4 Reagent Kit. The data were processed by a Multidimensional Scaling Analysis and a Hierarchical Clustering Analysis by the Euclidean algorithm with Complete Linkage program.

**In vitro IKK assay**. After a treatment with a series of HOIPIN-1 concentrations for 30 min, A549 cells were stimulated with 1 ng/ml of IL-1β for 15 min. The cells were then lysed with buffer, containing 50 mM Tris-HCl (pH 7.5), 150 mM NaCl, and 1% Triton X-100 (w/v), and immunoprecipitated with an anti-NEMO antibody and Protein A Sepharose. After extensive washing, the beads were suspended in buffer containing 50 mM Tris-HCl (pH 7.5) and 5 mM MgCl$_2$. The immunopre-cipitates were incubated with 5 μg/ml of either WT or S32/36A-mutated GST-IκBα (1–54 aa), in a 20 μl reaction, containing 50 mM Tris-HCl (pH 7.5), 5 mM MgCl$_2$, and 1 mM ATP, at 30 °C for 30 min, followed by immunoblotting.

**Analyses of TNFR complexes I and II**. The TNFR complex I analysis was per-formed as described previously[58]. Briefly, after a treatment with a series of HOIPIN-1 concentrations for 30 min, A549 cells were stimulated for the indicated times with 1 μg/ml FLAG-tagged TNF-α, and lysed in 1 ml lysis buffer, containing 50 mM Tris-HCl, pH 7.5, 150 mM NaCl, 1% Triton X-100, and complete protease inhibitor cocktail, for 15 min on ice. The cell lysates were then immunoprecipitated with 15 μl anti-FLAG M2 beads (Sigma), overnight at 4 °C. For the TNFR complex II analysis, A549 cells were treated with 40 ng/ml TNF-α and 20 μg/ml CHX. HOIPIN-1 was added 30 min prior to the TNF-α- and CHX treatments. The cell lysates were then immunoprecipitated with an anti-caspase 8 antibody (Santa Cruz) and immunoblotted.

**qPCR**. Cell lysis, reverse transcription, and qPCR were performed with a SuperPrep Cell Lysis RT Kit for qPCR (TOYOBO) and Power SYBR Green PCR Master Mix (Life Technologies), according to the manufacturers' instructions. Quantitative real-time PCR was performed with a Step-One-Plus PCR system (Applied Bio-systems) by the ΔΔCT method. For analyses with mouse-derived immune cells, total mRNA was extracted with an RNeasy Mini Kit (Qiagen), and then transcribed to cDNA with ReverTra Ace qPCR RT Master Mix with gDNA Remover (TOYOBO), according to the manufacturers' instructions. Quantitative real-time PCR was performed with Power SYBR Green PCR Master Mix (Life Technologies)

and a Step-One-Plus PCR system by the ΔΔCT method. Moreover, for the analysis with psoriasis-model mice, total mRNA was extracted by the same method used for the mouse immune cells. Quantitative real-time PCR was performed with TaqMan Gene Expression Master Mix (Applied Biosystems) and a Step-One-Plus PCR system, using the following TaqMan Gene Expression Assays. Primers used for this study are listed in Supplementary Table 2.

**Immunofluorescence staining**. A549 cells were fixed in 4% paraformaldehyde, permeabilized with 0.1% Triton X-100/PBS, and incubated with an anti-p65 antibody (Cell Signaling), followed by an incubation with an Alexa546-conjugated anti-rabbit antibody (Thermo Fisher). Nuclei were stained with DAPI (DOJINDO). Images were acquired with a Leica TCS SP5 confocal microsystem.

**Luciferase assay**. Transfection experiments were performed using PEI (poly-ethylenimine). The pGL4.32[*luc2P*/NF-κB-RE/Hygro] vector (Promega), the ISRE-luciferase reporter (a kind gift from Dr. Koichi Nakajima, Osaka City Univ.), the pGL3-IFNβ-luciferase reporter (a kind gift from Dr. Osamu Takeuchi, Kyoto Univ.), or pGL4-IFIT1-pro, which was constructed as described[59], was co-transfected into HEK293T or MEF cells with the pRL-TK *Renilla* Luciferase control reporter vector (Promega)[60]. At 24 h after transfection, the cells were lysed and the luciferase activity was measured with a GloMax 20/20 luminometer (Promega), using the Dual-Luciferase Reporter Assay System (Promega).

**Sendai virus infection**. MEFs were pre-incubated with HOIPIN-1 or -8 for 30 min, and then infected with Sendai virus (SeV) strain Cantell at a CIU (cell infectious unit) of 10 for 1 h. The washed cells were then incubated with or without HOIPINs for 8 h.

**In vitro ubiquitination assay**. For the in vitro ubiquitination assay of petit-LUBAC, samples containing 4.8 μg/ml E1, 9 μg/ml UbcH5c, 16 μg/ml WT- or mutants of petit-LUBAC, and 25 μg/ml ubiquitin, in 20 mM Tris-HCl (pH 7.5), 5.3 mM MgCl$_2$, 2 mM ATP, 0.5 mM DTT, and 0.1% Triton X-100, were incubated at room temperature for 30 min. The reaction was terminated by the addition of 1/4 volume of 4× LDS sample buffer (Invitrogen). For the ubiquitin-thioester inter-mediate formation assay, the N-terminally His$_6$-tagged ubiquitin was used, and samples were electrophoresed under non-reducing conditions. For the LUBAC assay, E1, E2 (UbcH5c), and baculovirus-expressed recombinant HOIL-1L/HOIP[14] were mixed with ubiquitin, ATP, and various concentrations of HOIPIN-1, and incubated at 37 °C for 1 h. Similarly, the in vitro E3 activities of MBP-full-length cIAP2 and GST-HHARI (aa. 177–395) were assayed by incubations at 30 °C for 1 and 2 h, respectively. The assay with E6AP was performed with an E6AP Ubiquitin Ligase Kit (K-230, Boston Biochem), and the ubiquitination of S5a was assessed. The in vitro ubiquitination assay for parkin was performed as follows: 5 μg/ml E1, 10 μg/ml E2 (UbcH7), 50 μg/ml MBP-parkin (aa. 217–645), and 150 μg/ml ubi-quitin were incubated in buffer, containing 50 mM Tris-HCl (pH 7.5), 5 mM MgCl$_2$, 1 mM DTT, and 2 mM ATP, at 37 °C for 2 h. Samples were subjected to SDS-PAGE, followed by immunoblotting.

**Mass spectrometric analyses**. Equal volumes of the WT and mutants of Petit-LUBAC (1 mg/ml) and HOIPIN-1, dissolved at 1 mM in 20 mM Tris-HCl, pH 7.5, 10% DMSO, were mixed, and diluted fivefold with 20 mM Tris-HCl, pH 7.5. Samples were incubated at room temperature for 1 h, and then acidified with 0.1% trifluoroacetic acid (TFA). The samples were then desalted on a reverse-phase SPE Tip-column (ZipTipC4, Millipore, Japan), and analyzed in the linear mode of an ultraflex III MALDI-TOF mass spectrometer (Bruker Daltonics).

**Mass spectrometric quantification of ubiquitin chains**. Liquid chromatography–tandem mass spectrometry (LC-MS/MS) analyses were per-formed essentially as previously described[61], with some modifications. Analytes (20 μg of whole-cell lysates) were separated by SDS-PAGE and stained with Bio-Safe Coomassie (Bio-Rad). Gel regions > 75 kDa were excised, and in-gel trypsin digestion was performed by an incubation at 37 °C for 12–15 h with 20 ng/μl Trypsin Gold (Promega), in 50 mM ammonium bicarbonate (AMBC), 5% acet-onitrile (ACN), pH 8.0. After trypsin digestion, ubiquitin-absolute quantification (AQUA) peptides (15 fmol/injection) were added to the extracted peptides. The concentrated peptides were diluted with 20 μl of 0.1% TFA containing 0.05% H$_2$O$_2$, and incubated at 4 °C overnight.

For the MS analysis, an Easy nLC 1200 (Thermo Fisher Scientific) was connected online to an Orbitrap Fusion LUMOS (Thermo Fisher Scientific) with a nanoelectrospray ion source (Thermo Fisher Scientific). Peptides were loaded onto a C18 analytical column (Ionopticks, Aurora Series Emitter Column, AUR2-25075C18A 25 cm ×75 μm, 1.6 μm FSC C18 with nanoZero fitting) and separated using a 45 min gradient (solvent A, 0.1% FA; solvent B, 80% ACN/0.1% FA). The Orbitrap Fusion LUMOS instrument was operated in the targeted MS/MS mode by the Xcalibur software, and the peptides were fragmented by HCD (higher energy collisional dissociation) with a normalized collision energy of 28. The MS/MS resolution, target AGC values, and isolation windows were set to 30,000, 5E4, and 2.0$m/z$, respectively. Data were processed using the PinPoint software, version 1.3

(Thermo Fisher Scientific), and the peptide abundance was calculated based on the integrated area under the curve of the selected fragment ions. The fragment ions used for quantification were previously described[61].

**Preparation of HOIPIN-bound HOIP^RING2-LDD.** The gene encoding the RING2-LDD domain of human HOIP (residues 853–1072; HOIP^RING2-LDD) was cloned into the pCold-GST expression vector using a Gibson assembly cloning kit (New England Biolabs). The N-terminal GST-fused HOIP^RING2-LDD was overproduced in *E. coli* strain Rosetta (DE3) cells at 15 °C (Invitrogen). The cells were disrupted by sonication and purified by chromatography on a Glutathione Sepharose FF column (GE Healthcare) and a ResourceQ anion exchange column (GE Health-care). After the GST tag was cleaved by HRV3C protease, HOIP^RING2-LDD was passed through a Glutathione Sepharose FF column. The flow-through fraction was loaded onto a HiLoad 16/60 Superdex75 size-exclusion column (GE Healthcare), equilibrated with 10 mM Tris-HCl buffer (pH 8.0) containing 50 mM NaCl.

To prepare the HOIP^RING2-LDD–HOIPIN conjugate, HOIP^RING2-LDD (400 μM) and HOIPIN-1 (520 μM) or HOIPIN-8 (520 μM) were mixed in 10 mM Tris-HCl buffer (pH 8.0) containing 50 mM NaCl, and incubated at 4 °C overnight. The HOIP^RING2-LDD–HOIPIN conjugate was purified by chromatography on a Superdex75 10/300 GL size-exclusion column (GE Healthcare), equilibrated with 10 mM Tris-HCl buffer (pH 8.0) containing 50 mM NaCl. The purified HOIP^RING2-LDD–HOIPIN conjugate was concentrated to ~10 mg/ml, using an Amicon Ultra-15 10,000 MWCO filter (Millipore), and stored at −80 °C until use.

**Crystallization.** Initial crystallization screening was performed by the sitting drop vapor diffusion method at 20 °C, with a Mosquito liquid-handling robot (TTP Lab Tech). We tested about 500 conditions with crystallization reagent kits supplied by Hampton Research and Qiagen. Initial hits were further optimized. The best crystals of the HOIP^RING2-LDD–HOIPIN-1 conjugate were grown at 20 °C by the sitting drop vapor diffusion method. The protein solution (0.2 μl) was mixed with an equal volume of the precipitant solution, containing 100 mM HEPES-Na (pH 7.5), 200 mM NaCl, and 25% PEG 3350, and was allowed to equilibrate against 50 μl of the precipitant solution. The best crystals of the HOIP^RING2-LDD–HOIPIN-8 conjugate were also grown at 20 °C by the sitting drop vapor diffusion method. The protein solution (0.5 μl) was mixed with an equal volume of the precipitant solution, containing 200 mM KCl and 20% PEG 3350, and was allowed to equilibrate against 500 μl of the precipitant solution. For data collection, the crystals were transferred to a cryostabilizing solution, which was the precipitant solution containing 30% glycerol. Cryoprotected crystals were flash cooled in liquid N$_2$.

**Structure determination.** Diffraction data sets were collected at beamline BL41XU in SPring-8 (Hyogo, Japan) and processed with XDS[62] and the CCP4 program suite[63]. The HOIP^RING2-LDD–HOIPIN-1 and HOIP^RING2-LDD–HOIPIN-8 structures were determined by molecular replacement, using Molrep[64]. The crystal structure of HOIP^RING2-LDD (PDB 4LJQ)[10] was used as the search model. The atomic models were corrected using Coot[65] with careful inspection, and refined using Phenix[66]. The final models were obtained after iterative correction and refinement of the models. All molecular graphics were prepared using PyMOL (DeLano Scientific; http://www.pymol.org).

**Cell survival assay.** The number of viable cells was measured with a CellTiter-Glo Luminescent Cell Viability Assay (Promega), which quantifies the ATP content, a Cell Counting Kit-F (DOJINDO) based on the degradation of Calcein-AM, and a trypan blue exclusion assay. For quantifying cellular cytotoxicity, a Cytotoxicity LDH Assay Kit-WST (DOJINDO), which is based on the lactate dehydrogenase (LDH)-release from damaged cells to the media, was used. Cytoplasmic histone-associated DNA fragments (mono- and oligonucleosomes), generated after induced cell death, were measured by a cell death detection ELISA (Roche), according to the manufacturer's instructions.

**Treatment of imiquimod-induced psoriasis-model mice.** Imiquimod-induced psoriasis-model mice were prepared as described[48]. Briefly, 8-week old female BALB/c mice, purchased from CLEA Japan, received a daily topical dose of 5% imiquimod cream (Beselna Cream from Mochida Pharmaceutical) on the shaved back skin for 7 days. Clobetasol propionate (0.05% Dermovate®, GlaxoSmithKline) or HOIPIN-1, initially dissolved in DMSO and mixed with Vaseline to a final concentration of 0.1% (w/w), was additively administered with the imiquimod cream. Control mice were treated with a vehicle cream (Vaseline Lanette cream, Fagron). Erythema, thickness, and scaling were evaluated, using a scale from 0 to 4 (ref. [48]). Mouse skin samples were fixed with 4% paraformaldehyde and embedded in paraffin, and sections were stained with hematoxylin and eosin. The thickness of the epidermis was analyzed with a BZ-8000 microscope (Biozero, Keyence, Osaka, Japan). Ki-67 and TUNEL stainings were performed as described[67].

**Statistics and reproducibility.** Data are shown as means ± SEM from the at least triplicated experiments. One-way ANOVA followed by a post hoc Tukey HSD test or Student's *t*-test was performed, using the KaleidaGraph software. For all tests, a *P* value of less than 0.05 was considered statistically significant.

**Ethics statement.** The protocols were approved by the Safety Committee for Recombinant DNA Experiments, the Safety Committee for Bio-Safety Level 2 (BSL-2) Experiments, and the Animal Experiment Committee of Osaka City University. Mice were euthanized according to guidelines, and all efforts were made to minimize the suffering of the animals.

**Reporting summary.** Further information on research design is available in the Nature Research Reporting Summary linked to this article.

## Data availability
The coordinates and structure factors for X-ray crystal structures have been deposited in the Protein Data Bank under the accession codes 6KC5 and 6KC6. RNA sequencing data has been deposited in SRA database under BioProject ID PRJDB9322. Supplementary Data 1 contains the source data underlying the graphs and charts presented in the article. Full blots are shown in Supplementary Information. All other data are available from the corresponding author on reasonable request.

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

## Acknowledgements

We thank Drs. Katrin Rittinger and Sandra Kümper (The Francis Crick Institute) for valuable discussions, Drs. Hitoshi Ohno (Tenri Hospital and Tenri Institute of Medical Research) and Masayuki Hino (Osaka City University) for the DLBCL cells, Drs. Koichi Nakajima (Osaka City University) and Osamu Takeuchi (Kyoto University) for luciferase reporters, Dr. Hiroyasu Nakano (Toho University) for helpful discussion, Ms. Wakaba Koeda and Ms. Yoshiko Fujita for technical assistance, and the Research Support Platform of Osaka City University Graduate School of Medicine for technical support. This work was partly supported by MEXT/JSPS KAKENHI grants (nos. JP16H06575, JP18H02619, and JP19K22541 to F.T., JP16H04750 to Y. Sato, JP18H05498 to F.O. and Y. Saeki, JP18H05501 to S.F., and JP18K06967 and JP19H05296 to D.O.), Osaka City University Strategic Research Grant 2017 for top priority research (to F.T.), Takeda Science Foundation (to F.T.), a Grant for Research Program on Hepatitis from the Japan Agency for Medical Research and Development (AMED—19fk0210050h0001 to F.T.), GSK Japan Research Grant 2017 (to D.O.), and a grant from the Nakatomi Foundation (to D.O.).

## Author contributions

D.O. and S.T. performed the cell biological experiments. Y. Sato performed the X-ray crystallographic analysis. K.H. performed MS and biochemical analyses. F.O. and Y. Saeki performed ubiquitin linkage analyses by MS. K.K., K.S., Y.M., and H.T.P. analyzed psoriasis-model mice. K.I. and S.O. synthesized HOIPINs. M.F. and T.I. contributed to the SeV and IRF3-5D analyses. D.T., S.S., K.T., Y. Saeki, S.F., and F.T. coordinated the study, and D.O., Y. Sato, F.O., K.H., K.S., D.T., S.F., and F.T. wrote the manuscript. All authors discussed the results and commented on the manuscript.

## Competing interests

The authors declare no competing interests.
