## [Peer Review File · Communications Biology]

Reviewers' comments:

Reviewer #1 (Remarks to the Author):

The linear ubiquitin chain assembly complex (LUBAC) is a trimeric E3 ubiquitin ligase complex responsible for the production of linear (also known as Met1-linked) ubiquitin chains in response to various cellular stimuli. The formation of these chains is critical for the activation of the canonical I κ B kinase (IKK) complex, which controls transcription factors such as NF- κ B and hence inhibits TNF-induced apoptosis. The authors have previously identified a compound called JTP-0819958 (subsequently renamed HOIPIN-1) that inhibits LUBAC activity in vitro (Katsuya et al. 2018, *SLAS Discov* 23, 1018-1029), as well as a subsequent derivative termed HOIPIN-8 (Katsuya et al. 2019, *Biochem Biophys Res Commun* 509, 700-706). In the current manuscript the authors show that both HOIPIN-1 and HOIPIN-8 inactivate LUBAC by forming a covalent bond with the catalytically important residue Cys885 in the HOIP subunit of the complex, as well as additional non-covalent interactions with the unique LDD domain in the C-terminus of HOIP. They report that HOIPIN-1 has little or no effect on several other E3 ligases tested. The remainder of the paper examines the effects of HOIPIN-1 or HOIPIN-8 on cellular signalling and cytokine production in a variety of cell-lines and innate immune signalling pathways following stimulation by agonists previously reported to trigger LUBAC activity, such as IL-1 β , TNF α , LPS, Poly(I:C), CD40L and CpG. They report that the HOIPINs accelerate the TNF α -induced extrinsic apoptotic pathway and induce cell death in activated B cell-like diffuse large B cell lymphoma cells. Finally, the authors report that HOIPIN-1 alleviates psoriasis-like symptoms in an imiquimod-induced mouse model system.

On the whole, the manuscript is well written and the experiments are well explained, with clear information on concentrations and timings for each stimulation and inhibitor treatment (some exceptions to this are noted below under minor comments). That said, readability is somewhat compromised by the authors' rather arbitrary choices regarding which of their two inhibitors to use, leading to a fragmented and unfocused experience for the reader (see major comment 6 below).

With the authors already having published two papers on the discovery of the HOIPIN compounds and their ability to inhibit linear ubiquitin chain formation in cell-free and cell-based assays, the current manuscript is a largely descriptive study that, in the opinion of this reviewer, falls short in its aim of establishing the HOIPINs as specific LUBAC inhibitors. The authors conclude that because all the effects observed upon HOIPIN treatment are those that they would expect from a LUBAC inhibitor that they have therefore been brought about by the HOIPINs' inhibition of LUBAC and that their compounds are therefore specific LUBAC inhibitors. But the authors have failed to carry out some of the basic controls necessary to make this claim and other modes of action are possible.

Major comments:

1. The major issue here is one of specificity. Are the HOIPINs bringing about their effects through the inhibition of LUBAC or for some other reason? One key measure of LUBAC activity is the production of linear ubiquitin chains following innate immune pathway stimulation, which should be suppressed following HOIPIN treatment. The authors do show this data following IL-1 β stimulation (Figure 1b) and also show reduced basal linear ubiquitin levels in unstimulated HBL1 cells (Figs 6g and 6f), but much of the paper concerns other signalling pathways, and for none of these are equivalent data shown. Such a control should have been one of the first blots performed for every experiment - it should be shown for each of the pathways that the authors have investigated that linear ubiquitin levels are suppressed by the HOIPINs - and their omission is a major concern. Admittedly, linear ubiquitin chain formation can be hard to detect after weak stimuli such as Poly(I:C), but there are methods available to compensate for this, such as enrichment with Met1-specific ubiquitin binders (see, for example, Emmerich et al. 2016, *Biochem Biophys Res Commun* 474, 452-461).

2. The ligand-induced formation of linear ubiquitin chains requires LUBAC recruitment to receptor signalling complexes, and is facilitated by the prior synthesis of Lys63-linked ubiquitin chains on complex components. All of the results reported by Oikawa et al. could be brought about by a failure to synthesise Lys63-linked ubiquitin after HOIPIN treatment. Disappointingly, the authors have not provided any evidence that Lys63-linked ubiquitin chain formation is normal in cells treated with their inhibitors.

3. As detailed above, the effects of HOIPIN-1 and HOIPIN-8 can potentially be explained by effects on any signalling components that lie "upstream" of IKK activation, which is the first event that the authors have measured in the majority of their experiments (the authors do examine RIP1 ubiquitylation and LUBAC recruitment to the TNFR signalling complex in Fig 1c, but the blots are so poor that it is difficult to conclude anything from them and their interpretation is complicated by the fact that RIP1 is reported to be modified with linear ubiquitin chains after 10 min TNF stimulation as opposed to the strictly lysine-linked ubiquitylation that the authors interpretation of the RIP1 ubiquitylation figure seems to require). The synthesis of Lys63-linked ubiquitin chains leads to the recruitment and activation of the protein kinase TAK1, which phosphorylates and activates IKK in IL-1, TNF and LPS signalling. Inhibition of TAK1 could therefore also bring about the phenotypes described by the authors. Oikawa and colleagues do show normal JNK phosphorylation after IL-1 and TNF treatment (Figs 1b and S2A) but this is somewhat downstream of TAK1 activation and a more direct measure would be to blot for TAK1 autophosphorylation, for which commercial antibodies are readily available.

4. If HOIPIN-1 is truly a specific inhibitor, it should have no effect in HOIP knockout cells, yet high concentrations of HOIPIN-1 reduce the phosphorylation of p105, p65 and I κ B α (Fig S2G). This suggests that the HOIPINs might have other targets within the cell that also affect the pathways investigated.

5. The authors suggest that Fig 3h shows HOIPIN-1 inhibiting the formation of a thioester-linked ubiquitin-HOIP reaction intermediate. I believe that the authors are probably correct, however, they have failed to show that these faint bands are sensitive to reducing agents such as DTT and as such they cannot claim with certainty that these bands are thioester-linked ubiquitin-HOIP conjugates (they could be mono-ubiquitylated HOIP).

6. The paper suffers from a lack of consistency regarding which inhibitor is used. Some experiments use HOIPIN-1 (the weaker of the two inhibitors), others use HOIPIN-8 and some are performed with both. This creates an unsatisfying experience for the reader and makes comparisons between experiments difficult as the two compounds are effective at different concentrations and may have distinct off-target effects. As a case in point, in Figure 3e the authors provide evidence that HOIPIN-1, when used at a concentration of 100 μ M, partially inhibits MBP-Parkin autoubiquitylation in vitro. They follow this up by looking at the ubiquitylation of the Parkin substrate MFN2 in the presence of a lower concentration (30 μ M) of a different inhibitor (HOIPIN-8) in cells (Fig 3f). No evidence is provided for HOIPIN-8's effect on MBP-Parkin in vitro, nor is the effect of HOIPIN-1 on MFN2 ubiquitylation investigated. It would seem prudent to check these things, especially since selectivity of the HOIPINs for HOIP rather than other RBR ligases is an important consideration.

7. In order to allow comparison of protein levels and phosphorylation states in extracts from the ABC-DLBCL and GCB-DLBCL cells in Figure 6 they should be run on the same gel. They are not, making comparisons difficult, if not downright impossible. Furthermore, contrary to what the authors state on page 24, in Figure 6c there does in fact seem to be enhanced PARP cleavage in the GCB-DLBCL cells when HOIPIN-1 is added, but the blot is less developed than that for the ABC-DLBCL cells.

Minor comments:

8. In Figure 2e, it is unclear what the stimulation is. The main text (page 12) seems to suggest Poly(I:C) but the figure legend states LPS.

9. There appears to be a typo on the y-axis of the right-hand graph of Figure 2e. ICxcl10 should read Cxcl10?

10. The cell type used in Figure 2f is not clear. The main text on page 12 says BMDM but the figure legend states MEFs. The authors should clarify.
11. In contrast to what is written in the main text (Page 12), which states that WT-MEF cells were used, the figure legend for Fig 2h states that the cells used were HEK293s. This, however, does not seem credible, as HEK293 cells do not normally express TLR3, which must instead be exogenously expressed to allow them to respond to Poly(I:C). Indeed, Figure 2h seems superfluous, as the same data has already been shown in Fig 2d.
12. Three quite different LPS concentrations are used across figures 2 and S4 (20 µg/ml in Figs 2a and S4C, 1 µg/ml in Fig S4E, and 100 ng/ml in Figs 2b, 2c, 2e, and S4D). Perhaps the authors could explain why different concentrations were used?
13. Should the figure legend for Figure 2e indicate Poly(I:C) rather than LPS stimulation? If it is LPS, then two thirds of the figure is a repetition of the data already presented in Figure 2b.
14. It would be helpful to know what cell type is used in Supplementary Fig 2C.
15. The HOIPIN-1 concentration used in Figure 3h is not stated.
16. The authors should consider defining in the figure legend what CBB stands for when written under Figure 3h.
17. On page 23, although the ABC part of ABC-DLBCL was defined in the abstract, it has not been defined in the main text, and should be defined here on first use.
18. On page 26, it would be helpful if the authors explained their reasons for staining for Ki-67 (a few words for those not familiar with this marker).
19. RNR should read RBR on page 27.
20. In Supplementary Table 2, should SudPHL4 read SU-DHL-4?
21. Although I hope it is a result of a low-res version of the paper being used for review, the quality of some figures is quite poor. For example, the sections in Supplementary Figure 10 are very hard to see, and blots such as the P-IKKα/β blot in Supplementary Fig 2A (which is very nearly completely blank) or the linear ubiquitin blot in Figure 1b (a key result) are of such poor quality that it is hard to trust the result that they appear to show.

Reviewer #2 (Remarks to the Author):

Summary of the manuscript

The authors of this paper have previously identified a group of small-molecule chemical inhibitors of LUBAC; a multi-protein complex known to be involved in several important cellular signalling pathways. The authors previously termed these inhibitors HOIPINs and identified several derivatives that inhibit NF-κB signalling with varying potency. Of these derivatives, HOIPIN-1 and -8 are the most potent LUBAC inhibitors reported and thus are the central focus of this study.

Here, Oikawa et al. have provided molecular insights into the mechanism of LUBAC inhibition by both HOIPIN-1 and -8. The authors clearly demonstrate that HOIPINs inhibit NF-κB, type I IFN and anti-apoptosis signalling pathways using an array of in vitro assays. Furthermore, the structures of HOIPIN-1 and -8 in combination with their target LUBAC subunits were solved by crystallography. Lastly, the authors provide preliminary evidence that HOIPIN's are effective at reducing the pathology associated with psoriasis in a mice model. From this, the authors postulate that HOIPINs may have therapeutic potential for the treatment of similar chronic inflammatory diseases and should be explored further.

Impression of the work

Overall the data are clearly presented and the experiments are mostly well-conducted, however there is limited novelty in the authors' findings. Firstly, LUBAC inhibitors have already been described in the literature (refs 21-27) and most importantly, a study published earlier this year solved the crystal structure of the interactions between LUBAC and an different inhibitor (ref 27 - Johansson et al. 2019 J Am Chem Soc). The interaction of this inhibitor (compound [5]) with the HOIP subunit shows remarkable similarity to HOIPINs interacting with HOIP in that they both types of inhibitors bind to the catalytic Cys885 residue. However, the structural analysis in this study revealed that HOIPIN-1 and -8 interact with unique HOIP residues which differentiates them from the other described LUBAC inhibitions. However, it is not at all surprising that LUBAC inhibitors of different flavours have different protein interactions with LUBAC components. Lastly, the authors fail to provide a clear explanation of how this novel HOIPIN binding information will be beneficial to understanding LUBAC biology in general. In summary, no new inhibitors are described, no new pathways are shown to involve LUBAC and the structural analysis, although robust, yielded incremental findings.

Specific comments

1. Page 14. The authors mention that the LUBAC-IRF3 co-expression experiments in illustrate that LUBAC is crucial for type I IFN production produced by PAMPs. However, these experiments alone are not entirely convincing given that overexpression systems are likely to not recapitulate a true physiological setting. The authors should consider confirming the importance of LUBAC-induced type I IFN signalling in response to PAMPs by using LUBAC subunit KO mice.
2. The study would have been enhanced if mutagenesis was performed on the residues mentioned on page 17 including positions 935, 905, 922 and 887 which were shown via structural analysis to share interactions with HOIPINs. The authors only generated a single mutant from this group (residue 935) however, it would be beneficial to explore these other residues to determine if they are important to HOIPIN binding and activity.
3. On page 21 the authors state that the mechanism underlying HOIP specificity is completely different between HOIPINs and compound (5) to emphasize the novelty of this study. However, both compounds interact with HOIP Cys855, and the differences in additional interacting residues is expected given the different structures of the inhibitors. To further provide evidence of different binding pockets the authors could perform synergy/antagonism studies to confirm whether the interacting residues HOIPINs truly differs from other LUBAC inhibitors.
4. Figure 7. Can the authors discuss whether any side-effects were observed in the mouse model with HOIPINs? In table 2 of the supplementary the authors present half maximal cytotoxicity data for HOIPIN-1 and HOIPIN-8 in vitro with a wide range of values (9.6 to >100 μM). Does this toxicity apply to the animals with the topical dosage? Are there any visible side effects in mice?
5. Sup Fig 1. The data in panels B and C are misleading for bendamustine treatment. In panel A, the authors show that this compound has no toxicity at the max. dosage tested (100 μM), however when assessing the NF-kB inhibitory effect of this compound the full range of non-toxic concentrations was not tested. The cytotoxicity IC50 is >100 μM but the authors stopped the analysis in panels B and D at 10 μM . In order to appropriately determine the effectiveness of bendamustine in inhibiting NF-kB signalling, or lack thereof, the authors need to include additional data points in their analysis for panels B and D.
6. Sup Fig. 8. The authors solely look at the effects of HOIPINs on A549 and MEFs, however it is not

clear why they specifically chose those cell lines to showcase that HOIPINs induce apoptosis? The authors should explain why the other cell lines weren't tested.

7. Sup Table 2. Why are there cytotoxicity values missing for HOIPIN-8 against many of the cell lines? Have these been ignored by the authors?

8. The discussion lacks comparison between the potency of HOIPINs and other LUBAC inhibitors characterised previously in the literature. It is hard to judge how much better/worse HOIPINs are compared to other described LUBAC inhibitors in the literature.

Reviewer #3 (Remarks to the Author):

In the manuscript by Oikawa et al., the authors characterize the inhibitors of LUBAC, HOIPINs, that they have previously published. The authors provide structural analyses and molecular mechanism of the mode of action of HOIPINs and provide evidence of their activity *in vivo*.

The study is interesting but there are a number of points that would need clarification.

1) Does HOIPIN-1 or -8 affect the levels of other ubiquitin chain types?

2) Related to RNA seq data: are the genes affected by HOIPINs targets of NF- κ B? If not exclusively, then how do the authors explain this broad effect of the inhibitor on IL1 β - regulated genes? It would be also important to see the RNAseq data on TNF signaling since LUBAC is so important in this signaling pathway.

3) Related to figure 5, since LUBAC is able to inhibit both apoptosis and necroptosis *in vitro*, the authors should try different cell types to test the effect of HOIPINs on apoptosis and necroptosis (A549 cells do not express RIPK3). Since HOIP deletion or catalytic inactivation sensitizes cells to death, why is treatment with HOIPINs not cytotoxic (5a) even at high doses alone or in combination with TNF alone? It would also be important to compare the cell death data with the mutant HOIP-C885A.

4) Many authors have published that HOIP or sharpin deficiency sensitizes cells to TNF-induced cell death with no need to add CHX. Can the authors comment on the fact that this is not the case in their conditions?

5) Related to the IMQ-model the cell death data is not clear. The authors should provide clearer images and quantification of TUNEL staining and perform another cell death staining for confirmation of results. Also the authors should explain the fact that HOIPINs induce cell death *in vitro* but not *in vivo*, and more importantly in an inflammatory model. This is rather counterintuitive also given published evidence that LUBAC is so crucial to regulate cell death.

We would like to take this opportunity to express our thanks to the reviewers for their constructive and useful remarks. We greatly appreciate the reviewers' help and assistance, as their efforts have significantly improved the paper.

Reviewer #1:

Major comments:

1. The major issue here is one of specificity. Are the HOIPINs bringing about their effects through the inhibition of LUBAC or for some other reason? One key measure of LUBAC activity is the production of linear ubiquitin chains following innate immune pathway stimulation, which should be suppressed following HOIPIN treatment. The authors do show this data following IL-1 β stimulation (Figure 1b) and also show reduced basal linear ubiquitin levels in unstimulated HBL1 cells (Figs 6g and 6f), but much of the paper concerns other signalling pathways, and for none of these are equivalent data shown. Such a control should have been one of the first blots performed for every experiment - it should be shown for each of the pathways that the authors have investigated that linear ubiquitin levels are suppressed by the HOIPINs - and their omission is a major concern. Admittedly, linear ubiquitin chain formation can be hard to detect after weak stimuli such as Poly(I:C), but there are methods available to compensate for this, such as enrichment with Met1-specific ubiquitin binders (see, for example, Emmerich et al. 2016, Biochem Biophys Res Commun 474, 452-461).

We appreciate the thoughtful suggestions by the reviewer. To clarify whether HOIPINs specifically suppress LUBAC-mediated linear ubiquitination, we performed immunoblotting and mass spectrometry analyses. As shown in the new Fig. 3a, we detected a significant enhancement of linear ubiquitin in TNF- α - and IL-1 β -treated A549 cells, and poly(I:C)-treated bone marrow-derived macrophages (BMDM) by immunoblotting using an anti-linear ubiquitin antibody. The enhanced linear ubiquitination did not occur in the presence of 30 μ M HOIPIN-8, although HOIPIN-8 did not affect the intracellular amounts of the K63-, K48-, and pan-polyubiquitin chains. Furthermore, we quantitated the ubiquitin linkages by a liquid chromatography-tandem mass spectrometry analysis. As shown in the new Supplementary Fig. 7a, the MS analysis indicated that stimulations with TNF- α , IL-1 β , and poly(I:C) significantly enhanced the intracellular contents of linear ubiquitin. However, HOIPIN-8 specifically suppressed the linear ubiquitin, but showed no effects on the amounts of the K11-, K29-, K48-, and K63-linked ubiquitin chains, which are the predominant Lys-linked ubiquitin

linkages. These results strongly indicated that the HOIPINs show selectivity for linear ubiquitin, but not Lys-linked ubiquitin chains. We described these findings in the Results (p. 14-15). K63-linked ubiquitination is known to be involved in NF- κ B activation. However, we could not detect an increase in the intracellular rate of K63-linked ubiquitin chain formation after 1 h stimulation with TNF- α , IL-1 β , and poly(I:C). K63-linked ubiquitin, the second most abundant ubiquitin linkage, functions not only in NF- κ B activation, but also in DNA repair, intracellular trafficking, and so on. Therefore, the intracellular content of K63-linked ubiquitin may not be largely affected by stimulation, or it might be down-regulated by deubiquitinases after 1 h stimulation. We discussed these subjects on p. 31.

2. The ligand-induced formation of linear ubiquitin chains requires LUBAC recruitment to receptor signalling complexes, and is facilitated by the prior synthesis of Lys63-linked ubiquitin chains on complex components. All of the results reported by Oikawa et al. could be brought about by a failure to synthesise Lys63-linked ubiquitin after HOIPIN treatment. Disappointingly, the authors have not provided any evidence that Lys63-linked ubiquitin chain formation is normal in cells treated with their inhibitors.

As described above, under our conditions, HOIPIN-8 showed no effects on the K63-linked ubiquitin level after the stimulations with TNF- α , IL-1 β , and poly(I:C) in A549 cells and BMDM (Fig. 3a and Supplementary Fig. 7a).

3. As detailed above, the effects of HOIPIN-1 and HOIPIN-8 can potentially be explained by effects on any signalling components that lie “upstream” of IKK activation, which is the first event that the authors have measured in the majority of their experiments (the authors do examine RIP1 ubiquitylation and LUBAC recruitment to the TNFR signalling complex in Fig 1c, but the blots are so poor that it is difficult to conclude anything from them and their interpretation is complicated by the fact that RIP1 is reported to be modified with linear ubiquitin chains after 10 min TNF stimulation as opposed to the strictly lysine-linked ubiquitylation that the authors’ interpretation of the RIP1 ubiquitylation figure seems to require). The synthesis of Lys63-linked ubiquitin chains leads to the recruitment and activation of the protein kinase TAK1, which phosphorylates and activates IKK in IL-1, TNF and LPS signalling. Inhibition of TAK1 could therefore also bring about the phenotypes described by the authors. Oikawa and colleagues do show normal JNK phosphorylation after IL-1 and TNF treatment (Figs 1b and S2A) but this is somewhat downstream of TAK1 activation and a more direct

measure would be to blot for TAK1 autophosphorylation, for which commercial antibodies are readily available.

We thank the reviewer for the important comment. We showed that the linear ubiquitination levels in complex I and an endogenous substrate of NEMO were significantly reduced with HOIPIN-1 (Fig. 1c, d). In contrast, we determined that the TNF- α -induced K63-linked ubiquitination of RIP1 (new Supplementary Fig. 2d), and the IL-1 β -induced autophosphorylation of TAK1 (Thr184/187) (new Supplementary Fig. 2c) were not largely down-regulated by HOIPIN-1.

4. If HOIPIN-1 is truly a specific inhibitor, it should have no effect in HOIP knockout cells, yet high concentrations of HOIPIN-1 reduce the phosphorylation of p105, p65 and I κ B α (Fig S2G). This suggests that the HOIPINs might have other targets within the cell that also affect the pathways investigated.

We thank the reviewer for the critical comment. As we previously described in the Discussion, we did not deny the possibility that HOIPINs may react with not only LUBAC but also other off-targets, since it has a thiol-reactive α,β -unsaturated carbonyl group. In the revised manuscript, we examined the effects of HOIPIN-8 in *HOIP*-deficient Jurkat cells. As shown in the new Supplementary Fig. 3f, HOIPIN-8 dose-dependently suppressed the TNF- α -induced phosphorylation of p100, p65, and I κ B α in wild-type Jurkat cells. In contrast, the effects of HOIPIN-8 on the phosphorylation of p100, p65, and I κ B α were minimal in *HOIP*-deficient Jurkat cells. We now show that the TNF- α treatment of *HOIP*-KO Jurkat cells effectively induces cell death, which was not completely suppressed by inhibitors of caspases and RIP1 (new Supplementary Fig. 10d). Therefore, the aberrant cellular homeostasis in *HOIP*-KO cells may affect the inhibitory effect of HOIPINs on NF- κ B activation. Furthermore, we presented the structural basis for the specificity of HOIPINs on HOIP (Fig. 4 and Supplementary Fig. 8), and showed that HOIPIN-8 exhibits a high IC₅₀ value of 11 nM to petit-LUBAC. Collectively, these results indicate that LUBAC is a high-affinity target of HOIPINs. It is beyond the scope of this study to analyze comprehensively the intracellular interactors of HOIPINs by synthesizing new chemicals, which can capture the interactors.

5. The authors suggest that Fig 3h shows HOIPIN-1 inhibiting the formation of a thioester-linked ubiquitin-HOIP reaction intermediate. I believe that the authors are probably correct, however, they have failed to show that these faint bands are sensitive

to reducing agents such as DTT and as such they cannot claim with certainty that these bands are thioester-linked ubiquitin-HOIP conjugates (they could be mono-ubiquitylated HOIP).

Based on the reviewer's comment, we performed an *in vitro* ubiquitination assay in the presence and absence of DTT, using wild-type ubiquitin and the N-terminally blocked His-ubiquitin. As shown in the new Supplementary Fig. 7d, petit-LUBAC-mediated linear polyubiquitination using WT-ubiquitin was not affected in the presence of 50 mM DTT. In contrast, a large portion of the modified band, denoted as HOIP⁴⁷⁴⁻¹⁰⁷²~His-Ub, disappeared in the presence of DTT, suggesting that the band mostly represents the thioester-linked His-Ub~HOIP intermediate. In the presence of HOIPIN-1, the linear polyubiquitination by WT-ubiquitin and the formation of the thioester-linked His-Ub~HOIP intermediate were strongly inhibited. These results indicated that HOIPIN-1 suppresses the thioester-linked ubiquitination of HOIP.

*6. The paper suffers from a lack of consistency regarding which inhibitor is used. Some experiments use HOIPIN-1 (the weaker of the two inhibitors), others use HOIPIN-8 and some are performed with both. This creates an unsatisfying experience for the reader and makes comparisons between experiments difficult as the two compounds are effective at different concentrations and may have distinct off-target effects. As a case in point, in Figure 3e the authors provide evidence that HOIPIN-1, when used at a concentration of 100 μ M, partially inhibits MBP-Parkin autoubiquitylation *in vitro*. They follow this up by looking at the ubiquitylation of the Parkin substrate MFN2 in the presence of a lower concentration (30 μ M) of a different inhibitor (HOIPIN-8) in cells (Fig 3f). No evidence is provided for HOIPIN-8's effect on MBP-Parkin *in vitro*, nor is the effect of HOIPIN-1 on MFN2 ubiquitylation investigated. It would seem prudent to check these things, especially since selectivity of the HOIPINs for HOIP rather than other RBR ligases is an important consideration.*

To compare the effects of HOIPIN-1 and -8 on RBR-type E3s, and to remove the redundancy, we now present an *in vitro* ubiquitination assay by GST-HHARI (Fig. 3c) and an intracellular assay for the Parkin-mediated ubiquitination of MFN1 (Fig. 3d) in the revised manuscript.

7. In order to allow comparison of protein levels and phosphorylation states in extracts from the ABC-DLBCL and GCB-DLBCL cells in Figure 6 they should be run on the same

gel. They are not, making comparisons difficult, if not downright impossible. Furthermore, contrary to what the authors state on page 24, in Figure 6c there does in fact seem to be enhanced PARP cleavage in the GCB-DLBCL cells when HOIPIN-1 is added, but the blot is less developed than that for the ABC-DLBCL cells.

According to the reviewer's comment, we electrophoresed cell lysates from ABC- and GCB-DLBCL cells on the same gels (new Fig. 6c and d). The results revealed that the basal expression levels of PARP in GCB-DLBCL cells were higher than those in ABC-DLBCL cells (Fig. 6c, short exposure). Although a treatment with HOIPIN-8 enhanced the cleavage of PARP in ABC-DLBCL cell lines, the amounts of cleaved PARP in GCB-DLBCL cell lines, such as HT and SU-DHL-4 cells, were not changed in the presence of HOIPIN-8 (Fig. 6c, long exposure of PARP). Moreover, HOIPIN-8 induced the cleavage of caspase 3 in ABC-DLBCL cells. Thus, these results suggested that HOIPINs enhance cell death in ABC-DLBCL cells.

Minor comments:

We thank the reviewer for their careful reading and helpful corrections.

8. In Figure 2e, it is unclear what the stimulation is. The main text (page 12) seems to suggest Poly(I:C) but the figure legend states LPS.

It is poly(I:C), and we corrected the figure legend.

9. There appears to be a typo on the y-axis of the right-hand graph of Figure 2e. ICxcl10 should read Cxcl10?

We corrected the typo.

10. The cell type used in Figure 2f is not clear. The main text on page 12 says BMDM but the figure legend states MEFs. The authors should clarify.

We corrected that they are "MEFs".

11. In contrast to what is written in the main text (Page 12), which states that WT-MEF cells were used, the figure legend for Fig 2h states that the cells used were HEK293s. This, however, does not seem credible, as HEK293 cells do not normally express

TLR3, which must instead be exogenously expressed to allow them to respond to Poly(I:C). Indeed, Figure 2h seems superfluous, as the same data has already been shown in Fig 2d.

To remove the redundancy, we deleted the previous Fig. 2h in the revised manuscript.

12. Three quite different LPS concentrations are used across figures 2 and S4 (20 µg/ml in Figs 2a and S4C, 1 µg/ml in Fig S4E, and 100 ng/ml in Figs 2b, 2c, 2e, and S4D). Perhaps the authors could explain why different concentrations were used?

We titrated the appropriate LPS concentrations for cell type, incubation time, and analytical methods.

13. Should the figure legend for Figure 2e indicate Poly(I:C) rather than LPS stimulation? If it is LPS, then two thirds of the figure is a repetition of the data already presented in Figure 2b.

They are stimulated by poly(I:C), rather than LPS. Therefore, the data do not overlap with those in Fig. 2b.

14. It would be helpful to know what cell type is used in Supplementary Fig 2C.

They are parental and *OPTN*-deficient HeLa cells, as described in the new Supplementary Fig. 3a.

15. The HOIPIN-1 concentration used in Figure 3h is not stated.

We described that the HOIPIN-1 concentration was 30 µM in the figure legend of new Fig. 3f, which corresponds to the previous Fig. 3h.

16. The authors should consider defining in the figure legend what CBB stands for when written under Figure 3h.

We described it as “Coomassie Brilliant Blue (CBB)” in the figure legend of the new Fig. 3f and the new Supplementary Fig. 7d.

17. On page 23, although the ABC part of ABC-DLBCL was defined in the abstract, it has not been defined in the main text, and should be defined here on first use.

We defined ABC-DLBCL in the main text (p. 25) upon its first use.

18. On page 26, it would be helpful if the authors explained their reasons for staining for Ki-67 (a few words for those not familiar with this marker).

To explain Ki-67, we inserted the phrase “which are a marker of cell proliferation in the epidermis” on p. 28.

19. RNR should read RBR on page 27.

We corrected the typo. Thank you.

20. In Supplementary Table 2, should SudPHL4 read SU-DHL-4?

We unified the nomenclature as SU-DHL-4 in this manuscript.

21. Although I hope it is a result of a low-res version of the paper being used for review, the quality of some figures is quite poor. For example, the sections in Supplementary Figure 10 are very hard to see, and blots such as the P-IKK α / β blot in Supplementary Fig 2A (which is very nearly completely blank) or the linear ubiquitin blot in Figure 1b (a key result) are of such poor quality that it is hard to trust the result that they appear to show.

We apologize for the inconvenience. In the revised manuscript, we presented higher magnification images of the TUNEL-positive cells and indicated the TUNEL-positive cells by *arrows* in new Supplementary Fig. 12b and c. We also showed the long exposure images for the P-IKK α / β blot in Supplementary Fig. 2a and the linear ubiquitin blot in Fig. 1b. Moreover, the immunoblotting analysis shown in the new Fig. 3a clearly indicates the reduced linear ubiquitin level in the presence of HOIPIN-8 by immunoblotting.

Reviewer #2:

Specific comments

1. Page 14. The authors mention that the LUBAC-IRF3 co-expression experiments in illustrate that LUBAC is crucial for type I IFN production produced by PAMPs. However, these experiments alone are not entirely convincing given that overexpression systems are likely to not recapitulate a true physiological setting. The authors should consider confirming the importance of LUBAC-induced type I IFN signalling in response to PAMPs by using LUBAC subunit KO mice.

As suggested by the reviewer, it is important to examine type I IFN signaling in an *in vivo* mice model. However, *Hoip*-KO (Sasaki Y *et al. EMBO J* **32**, 2463, 2013; Peltzer N *et al. Cell Rep* **9**, 153, 2014) and *Hoil-1*-KO (Peltzer N *et al. Nature* **557**, 112, 2018) mice are embryonic lethal. *Sharpin*-deficient cpdm mice exhibit severe chronic proliferative dermatitis (Gijbels MJ *et al. Am J Pathol* **148**, 941, 1996, Tokunaga F. *et al. Nature*, **471**, 633, 2011), and abnormalities in the immune system, such as the lack of Peyer's patches (HogenEsch H. *et al. J Immunol* **162**, 3890, 1999). Moreover, IFN- γ and IFN- α affect the expression level of *Hoip* and the NF- κ B activity in *Sharpin*-deficient cpdm mice (Tamiya H. *et al. J Immunol* **192**, 3793, 2014). Therefore, it would be difficult to investigate the PAMPs-induced type I IFN antiviral signal pathway using LUBAC subunit KO mice. In this study, we used various cell lines including *HOIP*- and *SHARPIN*-deficient cells, and in a future study we will investigate the physiological function of LUBAC in type I IFN signaling in LUBAC subunit conditional KO-mice.

2. The study would have been enhanced if mutagenesis was performed on the residues mentioned on page 17 including positions 935, 905, 922 and 887 which were shown via structural analysis to share interactions with HOIPINs. The authors only generated a single mutant from this group (residue 935) however, it would be beneficial to explore these other residues to determine if they are important to HOIPIN binding and activity.

We thank the reviewer for their thoughtful comment. We constructed the H887R, H887K, F905A, L922A, R935K, R935Q, R935E, D936S, D936T, D936N, and D936K mutants of HOIP, and expressed them in HEK293T cells with HOIL-1L, SHARPIN, and an NF- κ B-luciferase reporter. As shown in the new Supplementary Fig. 8d, these HOIP mutants exhibited drastically reduced NF- κ B activities as compared to that of the wild-type (WT), suggesting that these residues are critical for the E3 activity of LUBAC. Since the F905A mutant showed reduced but significant (~16% of WT) NF- κ B activity, we examined the

effect of HOIPIN-8 using the HOIP-F905A mutant with HOIL-1L and SHARPIN. As shown in the new Fig. 4f, the LUBAC containing HOIP-F905A mutant was insensitive to HOIPIN-8, although the WT-HOIP induced NF- κ B activity was drastically suppressed by HOIPIN-8. These results clearly indicated that Phe905 in HOIP contributes to the HOIPIN-8-binding. We described these findings in the Results (p. 20-21).

3. On page 21 the authors state that the mechanism underlying HOIP specificity is completely different between HOIPINs and compound (5) to emphasize the novelty of this study. However, both compounds interact with HOIP Cys855, and the differences in additional interacting residues is expected given the different structures of the inhibitors. To further provide evidence of different binding pockets the authors could perform synergy/antagonism studies to confirm whether the interacting residues HOIPINs truly differs from other LUBAC inhibitors.

We are grateful for the reviewer's comment. It is important to analyze whether HOIPINs synergistically or antagonistically react with α,β -unsaturated methyl ester-containing compounds, such as compound [5], to HOIP Cys885. Unfortunately, however, the α,β -unsaturated methyl ester-containing compounds reported by Dr. Rittinger's group (Johansson H. *et al. J Am Chem Soc* **141**, 2703, 2019) are not commercially available at present. Therefore, in a future study we will explore the effects of HOIPINs with α,β -unsaturated methyl ester-containing compounds to develop a specific inhibitor for LUBAC.

Additional note: Very recently (on December 5), Dr. Rittinger's group reported the structure of HOIP-RING2-LDD domain with their compounds[2], [3], [4], and [5] (Tsai YI, *et al. Cell Chem Biol*, in press). Moreover, by synthesizing HOIPIN-8 themselves, they showed the structural comparison between their compounds and HOIPIN-8, and the conclusion seems to be identical with ours.

4. Figure 7. Can the authors discuss whether any side-effects were observed in the mouse model with HOIPINs? In table 2 of the supplementary the authors present half maximal cytotoxicity data for HOIPIN-1 and HOIPIN-8 in vitro with a wide range of values (9.6 to >100 μ M). Does this toxicity apply to the animals with the topical dosage? Are there any visible side effects in mice?

In the revised manuscript, we show a photo of mouse back skin and the PASI score treated with 0.1% HOIPIN-1 alone (new Fig. 7a and b). Treatment with HOIPIN-1 alone

showed no apparent side effects, such as erythema, thickness, and scaling. Furthermore, the epidermal thickness was statistically unchanged between control Vaseline or DMSO and 0.1% HOIPIN-1 alone (Fig. 7d). Collectively, these results suggested that HOIPIN-1 showed no apparent toxicity with topical dosage or side effects in mice. We stated these findings in the Results (p. 27).

5. Sup Fig 1. The data in panels B and C are misleading for bendamustine treatment. In panel A, the authors show that this compound has no toxicity at the max. dosage tested (100 μ M), however when assessing the NF- κ B inhibitory effect of this compound the full range of non-toxic concentrations was not tested. The cytotoxicity IC₅₀ is >100 μ M but the authors stopped the analysis in panels B and D at 10 μ M. In order to appropriately determine the effectiveness of bendamustine in inhibiting NF- κ B signalling, or lack thereof, the authors need to include additional data points in their analysis for panels B and D.

According to the reviewer's comment, we examined the effect of bendamustine at higher concentrations. As shown in the new Supplementary Fig. 1b and c, the LUBAC-induced NF- κ B activation and the intracellular amounts of linear ubiquitin were not significantly affected in the presence of 30 μ M and 100 μ M bendamustine.

6. Sup Fig. 8. The authors solely look at the effects of HOIPINs on A549 and MEFs, however it is not clear why they specifically chose those cell lines to showcase that HOIPINs induce apoptosis? The authors should explain why the other cell lines weren't tested.

In the previous manuscript, we used A549 cells and MEFs for the cell death assay, since we examined the effects of HOIPINs on NF- κ B activation in these cells. RIP3, a critical factor for necroptosis, is not expressed in A549 cells, but it is expressed in MEFs. Therefore, although apoptosis is detectable in A549 cells, we can assess both apoptosis and necroptosis in MEFs. Moreover, *Sharpin*-deficient *cpdm* MEFs reportedly show accelerated cell death by TNF- α . We also analyzed the effect of HOIPINs on the cell death in seven DLBCL cell lines, as shown in the previous Fig. 6c.

In the revised manuscript, we further analyzed the effect of HOIPIN-8 on the cell death in parental HeLa, *HOIP*-deficient HeLa, HOIP-C885A mutant restored HeLa, wild-type MEF, *Sharpin*-deficient *cpdm* MEF, parental Jurkat, and *HOIP*-deficient Jurkat cells (new Supplementary Fig. 10). Since HeLa cells do not express RIP3, we examined the effect

of HOIPIN-8 on TNF- α +cycloheximide-induced apoptosis. In contrast, MEF and Jurkat cells express RIP3, and we analyzed TNF- α alone-induced apoptosis and necroptosis. Furthermore, the effects of the genetic ablation of LUBAC subunits on cell death were investigated in these cells, which revealed that the LUBAC activity is indispensable to suppress apoptosis and necroptosis. We believe that these analyses in various types of cells sufficiently support the effects of HOIPINs on cell death.

7. Sup Table 2. Why are there cytotoxicity values missing for HOIPIN-8 against many of the cell lines? Have these been ignored by the authors?

As depicted in the new Supplementary Table 2, we showed the IC₅₀ values of HOIPIN-8 for all of the cell lines tested with HOIPIN-1.

8. The discussion lacks comparison between the potency of HOIPINs and other LUBAC inhibitors characterised previously in the literature. It is hard to judge how much better/worse HOIPINs are compared to other described LUBAC inhibitors in the literature.

We appreciate the reviewer's comment. We described the IC₅₀ values and specificities of the reported LUBAC inhibitors in the Discussion (p. 29-30), and mentioned that HOIPIN-8 is the most potent LUBAC inhibitor.

Reviewer #3 (Remarks to the Author):

We greatly appreciate the thoughtful suggestions by the reviewer, especially on the cell death.

1) Does HOIPIN-1 or -8 affect the levels of other ubiquitin chain types?

In the revised manuscript, we performed immunoblotting and mass spectrometry analyses. As shown in the new Fig. 3a, we could detect the significant enhancement of the linear ubiquitin level in TNF- α - and IL-1 β -treated A549 cells, and poly(I:C)-treated bone marrow-derived macrophages (BMDM), by immunoblotting using an anti-linear ubiquitin antibody. In contrast, the enhancement was canceled in the presence of HOIPIN-8. Importantly, HOIPIN-8 did not affect the intracellular amounts of the K63-,

K48-, and pan-polyubiquitin chains. Furthermore, we quantitated the ubiquitin linkages by liquid chromatography-tandem mass spec analyses. As shown in the new Supplementary Fig. 7a, the replicated quantitative mass spectrometric analysis indicated that stimulations with TNF- α , IL-1 β , and poly(I:C) significantly enhanced the intracellular contents of linear ubiquitin. HOIPIN-8 specifically suppressed the linear ubiquitin content, whereas it showed no effects on the amounts of the K11-, K29-, K48-, and K63-linked ubiquitin chains, which are the predominant Lys-linked ubiquitin linkages. These results strongly indicated that HOIPIN-8 shows selectivity for linear ubiquitin, but not Lys-linked ubiquitin chains.

2) Related to RNA seq data: are the genes affected by HOIPINs targets of NF- κ B? If not exclusively, then how do the authors explain this broad effect of the inhibitor on IL1 β -regulated genes? It would be also important to see the RNAseq data on TNF signaling since LUBAC is so important in this signaling pathway.

RNA-seq is a transcriptome-wide comprehensive analysis. Therefore, in IL-1 β stimulated A549 cells (control experiments), 2,295 genes were up-regulated, and 2,247 genes were down-regulated. Moreover, the KEGG analysis suggested that multiple pathways, including TNF signaling, IL-17 signaling, apoptosis, pathogenic *E. coli* infection, metabolic, Parkinson disease, cellular senescence, gap junction, cancer, and so on, were directly or indirectly affected with IL-1 β stimulation. Thus, it is difficult to conclude by RNA-seq whether the IL-1 β -treatment exclusively affects NF- κ B targets, although the major NF- κ B targets were predominantly affected by HOIPINs, as shown in Fig. 1i. Furthermore, according to the reviewer's comment, we performed an RNA-seq analysis in TNF- α -treated A549 cells with or without HOIPIN-8 (new Supplementary Fig. 2f-h), and identified the down-regulation of major NF- κ B target genes. These results suggested that HOIPINs mainly affect NF- κ B target genes.

3) Related to figure 5, since LUBAC is able to inhibit both apoptosis and necroptosis in vitro, the authors should try different cell types to test the effect of HOIPINs on apoptosis and necroptosis (A549 cells do not express RIPK3). Since HOIP deletion or catalytic inactivation sensitizes cells to death, why is treatment with HOIPINs not cytotoxic (5a) even at high doses alone or in combination with TNF alone? It would also be important to compare the cell death data with the mutant HOIP-C885A.

4) Many authors have published that HOIP or sharpin deficiency sensitizes cells to TNF-induced cell death with no need to add CHX. Can the authors comment on the fact that this is not the case in their conditions?

We thank the reviewer for the critical comment. We first constructed *HOIP*-deficient HeLa cells and HOIP-C885A mutant-restored HeLa cells (new Supplementary Fig. 10a), and examined the TNF- α +CHX-induced cell death in the presence or absence of HOIPIN-8 (new Supplementary Fig. 10b). The results revealed that HOIPIN-8 induced apoptosis, but not necroptosis, in parental HeLa cells, since they lack RIP3 expression. However, the enhanced cell death by HOIPIN-8 was not detected in *HOIP*-deficient HeLa cells and C885A mutant-restored cells. Thus, the HOIPIN-8-mediated apoptosis is highly dependent on LUBAC.

To further investigate the involvement of necroptosis, we examined TNF- α alone-induced cell death in WT-MEF, *Sharpin*-deficient cpdm MEF, parental Jurkat, and *HOIP*-deficient Jurkat cells (Supplementary Fig. 10c, d). These cells are known to express RIP3. In wild-type MEFs, TNF- α -mediated necroptosis is specifically induced under the inhibition of LUBAC and apoptosis. In contrast, in *Sharpin*-deficient cpdm MEFs, TNF- α -mediated necroptosis was induced in the absence of HOIPIN-8, although the co-treatment with HOIPIN-8 and ZVAD further enhanced the cell death (Supplementary Fig. 10c).

In contrast to MEFs, the combined treatment with TNF- α and HOIPIN-8 induced significant cell death in parental Jurkat cells. Since both ZVAD and necrostatin-1 showed partial suppressive effects, apoptosis and necroptosis were simultaneously induced in Jurkat cells (Supplementary Fig. 10d). *HOIP*-deficient Jurkat cells were labile for TNF- α -induced cell death, and the co-treatment with TNF- α and HOIPIN-8 caused severe cell death, which was partially rescued by ZVAD and necrostatin-1. Collectively, these results indicated that the LUBAC activity is indispensable for the TNF- α -induced extrinsic apoptotic and necroptotic pathways. We described these findings in Results (p. 24-25). Although we anticipated that HOIPIN-8 has no enhancive effects on TNF- α alone-mediated cell death in *HOIP*-deficient Jurkat cells, it showed a significant increase in cell death. Since inhibitors of caspase and RIP1 did not fully rescue the cell death in *HOIP*-KO Jurkat cells, other pathways might be involved. Otherwise, unknown off-target(s) of HOIPIN-8 might affect the cell death in *HOIP*-KO Jurkat cells. This is an important issue to solve in the future, and we indeed plan to synthesize a novel compound(s) to comprehensively explore the interactors of HOIPINs by mass spec. Then, we will be able to identify the factor(s) involved in the cell death.

5) Related to the IMQ-model the cell death data is not clear. The authors should provide clearer images and quantification of TUNEL staining and perform another cell death staining for confirmation of results. Also the authors should explain the fact that HOIPINs induce cell death in vitro but not in vivo, and more importantly in an inflammatory model. This is rather counterintuitive also given published evidence that LUBAC is so crucial to regulate cell death.

In the revised manuscript, we presented higher magnification images of the TUNEL-positive cells, as shown in the new Supplementary Fig. 12c. Moreover, we counted the TUNEL-positive cells from sections of mouse back skin treated with IMQ and/or HOIPIN-1 (Supplementary Fig. 12d). Psoriasis is an inflammatory disorder, which is induced by an aberrant immune-inflammatory response in skin. Inflammatory cytokines and chemokines, in addition to dendritic cells, macrophages, Th1, Th17, and group 3 innate lymphoid cells (ILC3), are involved in the molecular pathogenesis of psoriasis. In the case of IMQ-induced psoriasis model mice, an increase in the TUNEL-positive apoptotic cells was not detected, suggesting that IMQ does not induce cell death. Since HOIPIN-1 significantly suppressed the IL-17, IL-22, and IL-23 levels (Fig. 7e), HOIPIN-1 seems to mainly have an anti-inflammatory effect on the activation of Th1, Th17, and ILC3, but does not cause cell death, followed by the inhibition of the feedback activation of immune cells. We described the major target of HOIPIN-1 during psoriasis in the Results (p. 28) and Discussion (p. 34).

Reviewers' comments:

Reviewer #1 (Remarks to the Author):

I would like to thank the authors for their efforts in addressing my previous concerns, the most pressing of which concerned the specificity of the HOIPIN compounds. To address this, the authors have analysed ubiquitin chain abundance in the presence and absence of HOIPINs after a number of stimulations. This analysis has been achieved by means of both western blotting and mass spectrometry. The authors have also sought to exclude TAK1 inhibition as a mechanism by which the HOIPINs may act by showing that the IL-1 β -induced phosphorylation of TAK1 is unaffected by HOIPIN-1 treatment. I am now much more confident that the reported phenotypes may be attributed to HOIP inhibition and feel that the manuscript is significantly improved as a result. Additionally, the authors have satisfactorily addressed my other concerns and the more minor points that I raised.

I do have a couple more (very minor) issues of wording to raise regarding this re-written version:

(i) In the new mass spec methods section the abbreviations AMBC and ACN and AQUA are used without definition. Although these will be familiar to many, and definitions can be found by referring to the previous publication referenced, it is probably best to define these for the reader.

(ii) The figure legend for Fig 1d talks about "Decreased linear ubiquitination of NEMO by HOIPIN-1". Obviously, HOIPIN-1 is not ubiquitylating NEMO, so this needs re-wording.

Reviewer #2 (Remarks to the Author):

Oikawa et al. have made changes to the manuscript which have improved it substantially.

This reviewer acknowledges that the authors have addressed all my comments to a satisfactory level in their rebuttal. In particular, the additional toxicity data and mutagenesis work has overall improved the clarity of the manuscript.

Although my stance on the novelty of this paper has not changed (as per the first round of reviews), I do believe that the work is scientifically sound and the experiments have been conducted well.

Reviewer #3 (Remarks to the Author):

In the revised manuscript by Oikawa and colleagues, the authors have improved the characterization of cell death and clarified most of my major concerns. However, I still have my doubts about the specificity of the inhibitor.

It is known that TNF, IL-1 β and poly(I:C) induce K63 and M1 chains in signaling complexes. In line with this, the authors demonstrate, by western blot and Mass Spectrometry, that cells stimulated with TNF, IL-1 β and poly(I:C) present with increased M1-Ub chains that are nicely reduced by treatment with HOIPIN-8 (Figure 3a and Supplementary Figure 7a). However, as the authors discuss, stimulation

with these ligands does not induce K63-Ub chains as they are present at the same extent as in control conditions. The authors state that HOIPIN-8 has no effect on these chains, which is actually true. However, unfortunately, since there is no stimulation-specific increase in K63- (and K48-) Ub chains in their conditions, this cannot be concluded. Judging by the levels of M1- versus K63-Ub chains, the fact that the authors do not see an induction on K63 Ub-chains might be due to the high total level of K63 in cellular lysates as compared to M1-Ub chains.

In order to claim specificity, the authors should find a condition in which they are able to see an increase in K63- (and other) Ub chains, and use this condition to study the effect of HOIPINs. For example, look at ubiquitination in respective signaling complexes rather than total cellular lysates.

Minor issues

1_ The authors clearly show that NF- κ B related genes are mainly affected by HOIPINs but they do not show other pathways regulated by M1-Ub chains (like cell death pathways) and what is the effect of HOIPINs therein (even if it is null). It would be good to have this information for robustness of the data.

2_ LUBAC deletion (or downregulation) by tamoxifen application in the skin resulted in mild dermatitis in adult mice (Taraborrelli et al., Nat Comm 2018). Similarly, topic application of Smac mimetics (a cIAP1/2 inhibitor) in the skin also resulted in dermatitis in mice (Anderton et al., J. Invest Dermatol 2017). Yet, HOIPINs do not have a similar effect, which would seem unexpected. Although the lack of toxicity of HOIPINs is encouraging for therapeutic purposes, it is a highly unexpected result which should be extensively investigated at this early stage of drug development. I understand that this may go beyond the scope of the present study but the authors should clearly and openly discuss this.

We thank for the reviewers for their careful reading and helpful comments.

Reviewer #1 (Remarks to the Author):

I do have a couple more (very minor) issues of wording to raise regarding this re-written version:

(i) In the new mass spec methods section the abbreviations AMBC and ACN and AQUA are used without definition. Although these will be familiar to many, and definitions can be found by referring to the previous publication referenced, it is probably best to define these for the reader.

We spelled out the abbreviations, such as ammonium bicarbonate (AMBC), acetonitrile (ACN), and ubiquitin-absolute quantification (AQUA), on page 50.

(ii) The figure legend for Fig 1d talks about “Decreased linear ubiquitination of NEMO by HOIPIN-1”. Obviously, HOIPIN-1 is not ubiquitylating NEMO, so this needs re-wording.

We changed the figure legend to “Suppression of IL-1 β -induced linear ubiquitination of NEMO in the presence of HOIPIN-1”, as shown on page 65.

Reviewer #3 (Remarks to the Author):

In the revised manuscript by Oikawa and colleagues, the authors have improved the characterization of cell death and clarified most of my major concerns. However, I still have my doubts about the specificity of the inhibitor.

It is known that TNF, IL-1 β and poly(I:C) induce K63 and M1 chains in signaling complexes. In line with this, the authors demonstrate, by western blot and Mass Spectrometry, that cells stimulated with TNF, IL-1 β and poly(I:C) present with increased M1-Ub chains that are nicely reduced by treatment with HOIPIN-8 (Figure 3a and Supplementary Figure 7a). However, as the authors discuss, stimulation with these ligands does not induce K63-Ub chains as they are present at the same extent as in control conditions. The authors state that HOIPIN-8 has no effect on these chains, which is actually true. However, unfortunately, since there is no stimulation-specific increase in K63- (and K48-) Ub chains in their conditions, this cannot be concluded. Judging by the levels of M1- versus K63-Ub chains, the fact that the authors do not see an induction on K63 Ub-chains might be due to the high total level of K63 in cellular lysates as compared to M1-Ub chains.

In order to claim specificity, the authors should find a condition in which they are able to see an increase in K63- (and other) Ub chains, and use this condition to study the effect of HOIPINs. For example, look at ubiquitination in respective signaling complexes rather than total cellular lysates.

We agree with the reviewer that it is important to investigate the effect of HOIPINs on Lys-linked ubiquitination. Among them, the K63 ubiquitin chain is another critical polyubiquitin chain for NF- κ B activation. In the previous revision, we examined the effect of HOIPIN-1 on the TNF- α -induced K63-ubiquitination of RIP1, a major target for K63-ubiquitination in TNF receptor signaling complex I, and it was not largely affected in the presence of HOIPIN-1 (Supplementary Fig. 2d and pages 8-9 in the revised manuscript). Moreover, the IL-1 β -induced activation of TAK1, a MAP3K with activation dependent on the K63-ubiquitin chain, was not suppressed by HOIPIN-1 (Supplementary Fig. 2e). Therefore, HOIPINs do not show any inhibitory effects on the TNF- α - and IL-1 β -mediated activation of JNK, a downstream MAPK for TAK1 (Fig. 1b, Supplementary Fig. 2a). In contrast, the IL-1 β -induced M1 ubiquitination of endogenous NEMO, a substrate for LUBAC, was drastically suppressed in the presence of HOIPIN-1 (Fig. 1d). Thus, we believe that we have already performed not only the global analysis of ubiquitin chains at the cell lysate level by immunoblotting and mass spec (Figure 3a and Supplementary Figure 7a), but also the examination of stimulation-specific targets for K63-ubiquitination in the NF- κ B activation pathway. We will be pleased to perform further analyses if necessary, but we hope that these results have been confirmed in the revised manuscript.

Minor issues

1_ The authors clearly show that NF- κ B related genes are mainly affected by HOIPINs but they do not show other pathways regulated by M1-Ub chains (like cell death pathways) and what is the effect of HOIPINs therein (even if it is null). It would be good to have this information for robustness of the data.

According to the reviewer's comment, we showed the data for the KEGG pathway analyses in the new Supplementary Fig. 2h-j. As pointed out by the reviewer, TNF- α -stimulation is involved in multiple pathways, including TNF signaling, inflammatory and infectious diseases, cytokine response, and necroptosis in A549 cells (Supplementary Fig. 2h). Most of these pathways seem to be related to the NF- κ B response. We concretely indicated the affected factors in the TNF signaling and necroptosis pathways with or without HOIPIN-8 (Supplementary Fig. 2i and j). On pages 9-10, we added the following sentences: "We

further examined the effect of HOIPIN-8 on TNF- α -induced gene expression by RNA-seq (Supplementary Fig. 2f-k). The TNF- α -treatment of A549 cells enhanced the expression of multiple genes involved in several pathways, affecting inflammatory signaling, infectious disorders, and cellular processes (Supplementary Fig. 2h). HOIPIN-8 showed suppressive effects on these pathways, but minimally affected the necroptosis pathway (Supplementary Fig. 2i-k).”

2_ LUBAC deletion (or downregulation) by tamoxifen application in the skin resulted in mild dermatitis in adult mice (Taraborrelli et al., Nat Comm 2018). Similarly, topic application of Smac mimetics (a cIAP1/2 inhibitor) in the skin also resulted in dermatitis in mice (Anderton et al., J. Invest Dermatol 2017). Yet, HOIPINs do not have a similar effect, which would seem unexpected. Although the lack of toxicity of HOIPINs is encouraging for therapeutic purposes, it is a highly unexpected result which should be extensively investigated at this early stage of drug development. I understand that this may go beyond the scope of the present study but the authors should clearly and openly discuss this.

As indicated by the reviewer, we added the following sentences on page 35 and cited the indicated references.

“It should be noted that the spontaneous mutation of *Sharpin* in mice (cpdm mice)^{15,19,40}, and the keratinocyte-specific deletion of *Hoip* or *Hoil-1*⁵⁵ induce severe dermatitis and cell death. Moreover, the epidermal-specific ablation of c-IAPs in mice, and the topical application of Smac mimetics, resulting in the depletion of c-IAPs and the reduced activation of LUBAC, also cause dermatitis⁵⁶. Since these results indicate that the excessive reduction of LUBAC activity causes skin lesions, careful titration of HOIPINs is necessary for the appropriate treatment of psoriasis.”

REVIEWERS' COMMENTS:

Reviewer #3 (Remarks to the Author):

The manuscript by Oikawa and colleagues has further improved and it is beautiful work. The specificity of HOIPINs remains somehow controversial but proper discussions by the authors would benefit the study.